©ⓒ Author(s) 2021. CC BY 4.0 License.





# Long-Term Atmospheric Emissions for the Coal Oil Point Natural Marine Hydrocarbon Seep Field, Offshore California

Ira Leifer[1], Christopher Melton[1], Donald R. Blake[2]

[1]Bubbleology Research International, Solvang, CA 93463, United States

[2]University of California, Irvine, Irvine, CA 92697, United States

*Correspondence to*: Ira Leifer (Ira.Leifer@bubbleology.com)

**Abstract.** In this study, we present a novel approach for assessing nearshore seepage atmospheric emissions through modeling of air quality station data, specifically, a Gaussian plume inversion model. Three decades of air quality station meteorology and total hydrocarbon concentration, *THC*, data were analysed to study emissions from the Coal Oil Point marine seep field offshore California. *THC* in the seep field directions was significantly elevated and Gaussian with respect to wind direction, $\theta$. An inversion model of the seep field anomaly, *THC'*($\theta$), derived atmospheric emissions. The model inversion is for the far field, which was satisfied by gridding the sonar seepage and treating each grid cell as a separate Gaussian plume. This assumption was validated by offshore *in situ* offshore data that showed major seep area plumes were Gaussian. Plume air sample THC was 85% methane, $CH_4$, and 20% carbon dioxide, $CO_2$, similar to seabed composition, demonstrating efficient vertical plume transport of dissolved seep gases. Air samples also measured atmospheric alkane plume composition. The inversion model used observed winds and derived the three-decade-average (1990-2021) field-wide atmospheric emissions of $83,500 \pm 12,000$ $m^3$ THC $day^{-1}$. Based on a 50:50 air to seawater partitioning, this implies seabed emissions of 167,000 $m^3$ THC $dy^{-1}$. Based on atmospheric plume composition, $C_1$-$C_6$ alkane emissions were 19, 1.3, 2.5, 2.2, 1.1, and 0.15 Gg $yr^{-1}$, respectively. The approach can be extended to derive emissions from other dispersed sources such as landfills, industrial sites, or terrestrial seepage if source locations are constrained spatially.

## 1 Introduction

### 1.1 Seepage and methane

On decadal timescales, the important greenhouse gas, methane, $CH_4$, affects atmospheric radiative balance far more strongly than carbon dioxide, $CO_2$ (IPCC, 2007, Fig. 2.21), yet $CH_4$ has large uncertainties for many sources (IPCC, 2013). Since pre-industrial times, $CH_4$ emissions have risen by a factor of ~2.5, and after stabilizing in the 1990s and early 2000s, resumed rapid growth since 2007 (Nisbet et al., 2019). The significantly shorter lifetime of $CH_4$ than $CO_2$ argues for $CH_4$ regulatory priority as emission reductions (and changes to the radiative balance) manifest more quickly as atmospheric concentrations decreases (Shindell, Faluvegi, Bell, & Schmidt, 2005). Further impetus for a $CH_4$ focus is a recent estimate that 40% $CH_4$ emissions reductions are feasible at no net cost for the oil and gas, O&G, industry (IEA, 2020), a major anthropogenic $CH_4$ source (IPCC, 2014). This is particularly salient given recent estimate that recent $CH_4$ increases are half from the O&G industry (Jackson et al., 2020).





For 2008-2017, global $CH_4$ top-down emissions estimates are 576 Tg yr$^{-1}$; 1 Tg=$10^{12}$ g, (550-594 Tg yr$^{-1}$) whereas
bottom-up approaches find 737 Tg yr$^{-1}$ (594-881 Tg yr$^{-1}$). Anthropogenic sources for 2008-2017 are estimated at 336-
376 Tg $CH_4$ yr$^{-1}$ based on bottom-up estimates. Natural sources include wildfires, wetlands, hydrates, and geological
seepage among others. Bottom-up estimates for natural sources are higher than top-down estimates including for
geological sources (Saunois et al., 2020). Geological sources (including seepage) are estimated at 63-80 Tg $CH_4$ yr$^{-1}$
with marine seepage estimated to contribute 20-30 Tg $CH_4$ yr$^{-1}$ (Etiope, Ciotoli, Schwietzke, & Schoell, 2019) or 5-
10 Tg $CH_4$ yr$^{-1}$ (Saunois et al., 2020). For comparison, marine non-geological $CH_4$ emissions are estimated at 4-10 Tg
yr$^{-1}$. The divergence of seepage emissions is based on uncertainty in the fraction of seabed emissions that reaches the
atmosphere and uncertainty in overall seabed emissions. Further complexity in assessing geological seepage $CH_4$
emissions arise because both seepage and O&G emissions source from the same geological reservoirs (Leifer, 2019)
and thus are isotopically similar (Schwietzke et al., 2016).

Seepage is where the migration of petroleum hydrocarbon gases and fluids in the lithosphere escape to the hydrosphere
and/or atmosphere from the reservoir formation where they have accumulated. The reservoir is sealed by a capping
layer to allow hydrocarbon accumulation. Thus, seepage requires a migration pathway through the capping layer
(Abrams, 2005), or the capping layer has been eroded away leaving a reservoir formation outcropping.

Marine seepage is widespread in every sea and ocean (Judd & Hovland, 2007). Quantitative seepage estimates (for
global budgets) are limited (though growing); see Leifer (2019) review and below for more recent. Fluxes for
individual marine seep vents and seep areas have been reported for the Gulf of Mexico (Johansen et al., 2020; Leifer
& MacDonald, 2003; Römer et al., 2019; T. C. Weber et al., 2014), the Black Sea (Greinert, McGinnis, Naudts, Linke,
& De Batist, 2010), the southern Baltic Sea (Heyer & Berger, 2000), various sectors of the North Sea (Borges,
Champenois, Gypens, Delille, & Harlay, 2016; Leifer, 2015; Römer et al., 2017), offshore Norway (Muyakshin &
Sauter, 2010; Sauter et al., 2006) and offshore Svalbard in the Norwegian Arctic (Veloso-Alarcón et al., 2019),
offshore Pakistan (Römer, Sahling, Pape, Bohrmann, & Spieß, 2012), the arctic Laptev Sea (Leifer, Chernykh,
Shakhova, & Semiletov, 2017), the East Siberian Arctic Sea (Shakhova et al., 2013), the South China Sea (Di, Feng,
Tao, & Chen, 2020), New Zealand's Hikurangi Margin (Higgs et al., 2019), the Cascadia Margin (Riedel et al., 2018),
and the Coal Oil Point (COP) marine hydrocarbon seep field, hereafter COP seep field, in the northern Santa Barbara
Channel, offshore Southern California (Hornafius, Quigley, & Luyendyk, 1999), and for numerous individual vents
in the field (Leifer, 2010).

Most seep emission estimates are from short-term field campaigns that provide a snapshot rather than annualized
values. Seep emissions vary on timescales from tidal (Leifer & Boles, 2005; Römer, Riedel, Scherwath, Heesemann,
& Spence, 2016) to seasonal (Bradley, Leifer, & Roberts, 2010) to decadal (Fischer, 1978; Leifer, 2019). Additional
temporal variability arises from transient emissions – pulses lasting seconds to minutes (Greinert, 2008; Schmale et
al., 2015) to decades (Leifer, 2019). This shortcoming is being addressed by benthic (seabed) observatories and cabled





observatories, e.g., Wiggins, Leifer, Linke, and Hildebrand (2015); Greinert (2008), (Kasaya et al., 2009; Römer et
al., 2016; Scherwath et al., 2019). Still, benthic observatories are costly and thus rare.

Seepage contributes to oceanographic budgets and to a lesser extent, atmospheric budgets due to water column losses
with significant uncertainty in the partitioning. As a result, uncertainty in the atmospheric contribution is much larger
than the (significant) uncertainty in seabed emissions. Seepage partitioning between the atmosphere and ocean - where
microbial degradation occurs on timescales inversely related to concentration (Reeburgh et al., 1991) - depends
primarily on depth (Leifer & Patro, 2002), with little to none of deepsea seabed emissions reaching the atmosphere,
e.g., (Römer et al., 2019). In contrast, very shallow seepage (meter scale) largely entirely reaches the atmosphere by
both direct bubble-meditated transfer and diffusive transport. For intermediate depths, the ocean/atmospheric
partitioning is complex and depends on depth, bubble flux, bubble size distribution, and bubble surface conditions,
among other characteristics (Leifer & Patro, 2002), whereas the indirect diffusive flux (proximate and distal) depends
on bubble dissolution depth (Leifer & Patro, 2002), turbulence vertical transport in the winter wave-mixed layer
(Rehder, Keir, Suess, & Rhein, 1999), microbial oxidation losses, and exchange through the sea-air interface.

A range of approaches have been used to estimate the sea-air flux. The most common is by measuring the atmospheric
and water concentrations and applying air-sea gas exchange theory for the measured wind speeds, e.g., Schmale,
Greinert, and Rehder (2005) for Black Sea seepage under weak wind speeds.

Sea-air exchange is a diffusive turbulence transfer process that depends on the air-sea concentration difference and
the piston velocity, $k_T$, which depends on gas physical properties, wind speed, $u$ (Liss & Duce, 2005), wave
development (Zhao, Toba, Suzuki, & Komori, 2003), wave breaking (Liss & Merlivat, 1986), and surfactant layers at
low wind speeds which suppress gas exchange (Frew et al., 2004). $k_T$ increases rapidly and non-linearly with $u$ and
has been parameterized as piecewise linear functions (Wanninkhof, Asher, Ho, Sweeney, & McGillis, 2009) or as a
cubic function (Nightingale et al., 2000). Air-sea gas exchange theory is for (relatively) homogeneous atmospheric
and oceanographic fields (concentrations, winds, wave development), and thus is inappropriate for point-source
(bubble-plume) emissions and for the near-field downcurrent plume, which tends to be heterogeneous.

An alternate approach is to derive atmospheric emissions by Gaussian plume inversion. Leifer, Luyendyk, Boles, and
Clark (2006) derived emissions for a blowout from Shane Seep in the COP seep field. This neglects of course, the
portion that dissolves in the ocean and drifts downcurrent out of the bubble plume's vicinity – the dissolved portion
that evades the ocean in the vicinity of the plume is counted by the inversion. Finally, seabed bubble size measurements
or an assumed bubble size distribution can be used to initialize a numerical bubble model to predict atmospheric direct
fluxes (Leifer et al., 2017; Schneider von Deimling et al., 2011). Additionally, a dispersive model for the fraction of
bubble-mediated transported gases to the wave mixed layer could then estimate the indirect diffusive atmospheric
emissions by turbulence transport and sea-air gas evasion. As noted, there are many variables, many of which may
not be measured, emphasizing the importance of model validation.




### 1.2 Study motivation

In this study, we present a novel approach for assessing nearshore seepage atmospheric emissions –air quality station
data modeling, specifically using a Gaussian plume inversion model. This model requires that source locations are
mapped, spatially stable, and lie within a fairly constrained distance range band. These conditions are met for the West
Campus air quality Station (WCS) and the nearby offshore Coal Oil Point (COP) marine hydrocarbon seep field. COP
seep field lies in shallow coastal waters of northern Santa Barbara Channel, CA. Spatial constraint is provided by
geological structures, such as faults, that constrain emission locations. The Gaussian plume model assumes the source
is in the far field, whereas WCS is in the nearfield for the extensive COP seep field. To satisfy the far field criterion,
the source was gridded and each grid cell's emissions treated as a distinct (distant) Gaussian plume. This
characterization was validated in an offshore survey of several focused COP seep field seepage areas, which were
well-modeled as Gaussian plumes.

Thus, this study demonstrates a novel approach to deriving emissions from an air quality station data for an area source
such as natural marine seepage. This approach could be used to derive emissions from other dispersed sources such
as landfills, industrial sites, or natural terrestrial seepage where the source locations can be constrained spatially.

### 1.3 Water column marine seabed seepage fate

Seep seabed $CH_4$ partitions between the atmosphere and water column depends on seabed depth and emission
character – as bubbles, bubble plumes (Leifer & Patro, 2002), or dissolved $CH_4$. Dissolved $CH_4$ migration through the
sediment is oxidized largely by near seabed microbes (Reeburgh, 2007), termed the microbial filter, negating its
contribution. Thus, $CH_4$ transport through the seabed's microbial filter is bubble-mediated.

As seep bubbles rise, they dissolve, losing gas to the surrounding water at a rate that decreases with time; smaller and
more soluble gases dissolve faster than larger and less soluble gases, i.e., fractionation (Leifer & Patro, 2002).
Additionally, larger bubbles transport their contents upwards more efficiently than smaller bubbles (Leifer et al.,
2006). Sufficiently large bubbles reach the sea surface with a significant fraction of their seabed $CH_4$ from depths of
even hundreds of meters (Solomon, Kastner, MacDonald, & Leifer, 2009). There are synergies, too with higher plume
fluxes driving a stronger upwelling flow that transports plume fluids with dissolved gases upwards towards the surface
where air-sea gas exchange drives evasion (Leifer, Jeuthe, Gjøsund, & Johansen, 2009). Another synergy is from
elevated dissolved plume $CH_4$ concentration (Leifer, 2010; Leifer et al., 2006), which slows dissolution. Also, bubbles
are oil-coated, which slows dissolution.

Moreover, gases in bubbles that dissolve in the wave mixed layer (or reaches it by the upwelling flow), then diffuse
by turbulence to the air-sea interface. Note, some of this dissolved $CH_4$ is removed by microbial degradation and thus





never reaches the air-sea interface. Thus, there are two timescales that govern the fraction that evades – the microbial
degradation timescale, which decreases as concentrations increase, and the diffusion timescale, which decreases with
increasing wind speed. As a result, there is a dissolved plume that drifts downcurrent, from which evasion creates a
linear-source atmospheric plume, with dissolved plume concentrations decreasing with time from sea-air gas exchange
losses, microbial oxidation, and diffusion.

**1.4 Atmospheric Gaussian plumes**
Strong focused atmospheric plumes are created from seep bubble bursting at the sea surface and dissolved gas evasion
within the bubble surfacing footprint, which is enhanced by water-side turbulence from rising and bursting bubbles
(Leifer et al., 2015). The atmospheric plume evolution is described by the Gaussian plume model (Hanna, Briggs, &
Hosker Jr., 1982), which relates downwind concentrations to wind transport and turbulence dispersion and is described
in **Supp. Sec. S1**.

**1.5 Setting**
**1.5.1 The Coal Oil Point seep field**
The COP seep field (**Fig. 1**) is one of the largest seep fields in the world, with estimated seabed emissions, $E_B$, for
1995-1996 of $1.5 \times 10^5 \pm 2 \times 10^4$ m$^3$ THC dy$^{-1}$ (Hornafius et al., 1999). Hereafter emissions and concentrations are for
total hydrocarbon, THC, unless noted. Of these seabed emissions, Clark, Washburn, Hornafius, and Luyendyk (2000)
estimated that half the COP seep field $E_B$ reach the atmosphere in the near field. This is due to shallowness, bubble
oiliness, high plume bubble densities, and turbulence mixing within the wave mixed layer.

Geological structures play a critical role in the spatial distribution of seepage (Leifer, Kamerling, Luyendyk, & Wilson,
2010), which lies along several trends in waters from a few meters to ~85 m deep. These trends follow geologic
structures including anticlines, synclines, and faults in the reservoir formation, the Monterey Formation. Faults provide
migration pathways with seepage scattered non-uniformly along the trends, including focused seep areas that are
highly active, localized, and often are associated with crossing faults and fractures (Leifer et al., 2010). Seepage in
these areas typically surrounds a focus and decreases with distance, primarily along linear trends (Leifer, Boles,
Luyendyk, & Clark, 2004). See **Supp. Table S3** for informal names and locations of selected focused seep areas.

**1.5.2 Coal Oil Point seep field emissions and composition**
COP seep field sources from the South Ellwood oil field whose primary source rock is Monterey Formation, which is
immature to marginally mature. Petroleum gases from marine organic materials have relatively higher proportion of
ethane, propane, butane, etc., relative to methane as compared to petroleum gases from terrestrial organic materials.
The wet gas fraction ($C_2$-$C_5$/$C_1$-$C_5$) indicates a thermogenic origin of greater than 0.05 (Abrams, 2017). Of the


saturated alkanes, the alkenes (olefins) are of biological origin. Additionally, the ethane/ethene ratio and
propane/propene ratios can be indicators of seep gas biogenic modification with values above 1000 indicating purely
thermogenic origin (Abrams, 2017; Bernard, Brooks, & Zumberge, 2001).

In this study, we analyse WCS (located at 34° 24.897'N, 119° 52.770'W) atmospheric THC. Clark, Washburn, and
Schwager (2010) report average seep field seabed $CH_4$, $CO_2$, and non-methane hydrocarbons (NMHC), of 76.7, 15.3,
and 7.7%, respectively, with Trilogy Seep seabed compositions of 67, 21, and 7.8%, respectively. With respect to
alkanes, seabed bubbles are 90.4% $CH_4$ and 8.6% NMHC. $CO_2$ rapidly escapes the bubbles and is negligible (<1%)
at the sea surface. At the sea surface, $CH_4$ in bubbles is ~90% with NMHC making up the remaining 10%, neglecting
air gases (Clark et al., 2010). Note, whereas seep THC is predominantly $CH_4$, THC from terrestrial directions arises
from NMHC from traffic and other anthropogenic sources as well as $CH_4$ from pipeline leaks, terrestrial seeps, etc.

**1.5.3 Northern Santa Barbara Channel climate**
Diurnal and seasonal wind cycles are important to the transport of the seep atmospheric emissions. The Santa Barbara
climate is Mediterranean with a dry season and a wet seasons when storms occur infrequently (Dorman & Winant,
2000). The semi-permanent eastern Pacific high-pressure system plays a dominant controlling role in weather in the
Santa Barbara coastal plain. This high-pressure system drives light winds and strong temperature inversions that act
as a lid that restricts convective mixing to lower altitudes. The coastal California boundary layer is shallow, 0 to 800
m (Edinger, 1959), generally 240-300 m around Santa Barbara (Dorman & Winant, 2000). Additionally, coastal
mountains provide physical barriers to transport (Lu, Turco, & Jacobson, 1997).

As a coastal environment, the land/sea breeze is important to overall wind flow patterns with weak offshore night
winds and stronger onshore afternoon winds (Dorman & Winant, 2000). In coastal Santa Barbara, warming on
mountaintops and more interior arid land relative to cooler marine temperatures drives the sea breeze. Downslope
nocturnal flows warm nighttime surface temperatures, moderating the coastal diurnal temperature cycle (Hughes, Hall,
& Fovell, 2007).

Typical morning winds are calm and offshore, often accompanied by a cloud-filled marine boundary layer, just 50–
150 m thick (Lu et al., 1997). The marine layer usually (but not always) "burns off" mid–morning after which
temperatures rise, the boundary layer thickens and winds shift clockwise from offshore to eventually prevailing
westerlies aligned with the coastal mountains. Mid-day through the afternoon, winds strengthen, often leading to
whitecapping that can continue into the evening before the boundary layer collapses and winds return to the nocturnal
pattern.

## 2 Methods

### 2.1 West Campus Station data

WCS data includes wind speed, $u$, direction, $\theta$, and THC concentration, $C$. Daily instrument calibration occurs after midnight, rendering $C$ unavailable 00:50 to 02:09 local time, LT. WCS was improved significantly in 2008 from 1-hour to 1-minute time resolution, which allowed far higher values of $C$ and $u$ due to the shorter averaging times. Data analysis uses custom routines as well as standard routines and functions in MATLAB (MathWorks, MA).

First, WCS data were quality controlled to remove all values of $C$ during the daily calibration, as well as to interpolate neighboring values that were unrealistically low, i.e., $C$ less than 1.6 ppm in the 1990s and 1.85 ppm in the 2000s. Data since 2008 were smoothed by nearest-neighbor averaging, yielding 3-minute time resolution. Data prior to 2008 were hourly and were not smoothed. Wind data were nearest-neighbor averaged after decomposing into north and east components, followed by recalculation of $u$ and $\theta$.

### 2.2 *In situ* marine surveys

Offshore *in situ* survey data were collected by the *F/V Double Bogey*, a 12-m, 9-ton, fishing vessel with a near waterline deck (~0.2 m) and low overall profile (cabin at ~2.2 m). A sonic anemometer (VMT700, Vaisala) was mounted on a 6.5 m, 5-cm (2") diameter aluminum mast and measured 3D winds. Continuous, 5 Hz $CH_4$ and $CO_2$ data were collected by a Cavity Enhanced Absorption Spectroscopy (CEAS) analyzer (FGGA, LGR Inc., San Jose, CA). Vessel location and time were from a Global Positioning System (GPS) at 1 Hz (19VX HVS, Garmin, KS). $CH_4$ and $CO_2$ calibration with a greenhouse gas air calibration standard ($CH_4$: 1.981 ppmv; $CO_2$: 404 ppmv, Scott Marin, CA, purchased 2015, Sigma Aldritch, St Louis, MO).

Data are real time integrated and visualized in Google Earth on a portable computer (Spectre360, HP) using custom software, written in MATLAB (MathWorks, MA) for AutoMObile trace Gas (AMOG) Surveyor, described elsewhere (Leifer, Melton, Fischer, et al., 2018; Leifer, Melton, Manish, & Leen, 2014; Leifer, Melton, Tratt, et al., 2018; Leifer et al., 2016). Real-time visualization facilitates adaptive surveys, wherein the survey route is modified based on real time data to improve outcomes (Thompson et al., 2015) - in this case to facilitate plume tracking and to ensure transects were near orthogonal to the wind.

Accurate, absolute winds are calculated from relative winds after accounting for vessel motion and filtering for non-physical velocity changes due to GPS uncertainty (Leifer, Melton, Fischer, et al., 2018). Filtering removes transient wind speeds that are not relevant to plume transport. The filter interpolates GPS positions flagged as unrealistic.

Whole air samples were collected in evacuated 2-liter stainless steel canisters, which were filled gently over ~1 minute from ~1 m above the sea surface. The filled canisters were analyzed in the Rowland/Blake laboratory at the University





246 of California, Irvine for carbon monoxide, CO, CH$_4$, and C$_2$-C$_7$ organic compounds. Samples were analysed by a gas

247 chromatography multi-column/detector analytical system utilizing flame ionization detection.

248

### 2.3 Seep plume emissions model

250 The plume inversion model is a three-step process (Leifer, Melton, Fischer, et al., 2018; Leifer, Melton, Tratt, et al.,

251 2018; Leifer et al., 2016). Emissions from focused seep areas were derived from offshore data by first fitting Gaussian

252 function(s) to orthogonal transect $C'$ data, termed the data model. $C'$ is relative to $C$ outside the plume, derived by

253 linear interpolation across the plume. The data model is derived by error minimization using a least-squares linear-

254 regression analysis (Curve fitting toolbox, MathWorks, MA). Next, the Gaussian plume model (**Eqn. S1; Supp. Figs.**

255 **S1; S2**) is fit to the data model. Transect data are collected close to orthogonal to the wind direction and are projected

256 in the wind direction onto an orthogonal plane. A validation study of the approach is described in Leifer, Melton,

257 Tratt, et al. (2018) where model-derived emissions were compared with remote sensing-derived emissions (which are

258 largely insensitive to transport). The study found *in situ* and remote-sensing derived emissions agreed within 11%.

259

### 2.4 Seep field emissions model

261 The seep plume emissions inversion model is based on gridding the seep field into numerous small additive Gaussian

262 plumes to represent the area emissions and was written in MATLAB (MathWorks, MA). This assumes that each sea-

263 surface grid cell contributes a Gaussian plume, an assumption that was tested with offshore survey data downwind of

264 several active seep areas.

265

266 The definition of area versus point source depends on the relevant length scales – an area source is well approximated

267 as a point-source plume if sufficiently downwind (far field), where the downwind distance depends on the source's

268 size scale and meteorological conditions. Whereas WCS is near field for the entire seep field plume, the small plumes

269 from each grid cell is in the far field for WCS.

270

271 The area source was based on a Sept. 2005 sonar survey sonar return, $\omega$, map (**Fig. 1**), see Leifer et al. (2010) for

272 sonar survey details. Simulations used sonar data gridded at a hybrid 22/56-m in a UTM coordinate system, with origin

273 at WCS. Specifically, gaps in the 22-m map were filled from the 56-m map (**Supp. Fig. S3**). The probability

274 distribution of $\omega$ was used to identify the noise level (**Supp. Fig. S4**) as in Leifer et al. (2010).

275

276 The model calculates a Gaussian plume for $E(i,j)$ for grid cells $i00$ and $j$, for each grid cell with $\omega$ above noise for the

277 observed $u(\theta)$ in the wind direction $\theta$ and a typical Santa Barbara channel boundary layer. The initial $E(i,j)$ is by

278 scaling such that the integrated sonar return ($\int\omega(x,y)$) scales to $E=1.5\times10^5$ m$^3$ dy$^{-1}$, i.e., $E_B$ from Hornafius et al. (1999).

279 The Gaussian plume is calculated in a Cartesian coordinate system (**Supp. Fig. S5A**) and then rotated to $\theta$ and linearly

280 interpolated to double the spatial resolution. The rotated plume then is regridded to UTM coordinates using the


ffgrid.m function (**Supp. Fig. S5B**). Interpolation helps prevent gaps in the regridded plume map. The regridded plume
is renormalized to ensure total mass is conserved before and after these operations. Rotated plumes are translated to
the seep field grid and added, yielding $C'_{Sim}(i,j)$, the simulated plume anomaly (**Supp. Fig. S5C**).

The model scans $\theta$ for the seep directions ($110°<\theta<330°$) and calculates the simulated plume anomaly, $C'_{Sim}(\theta)$ at
WCS, which is compared with the observed $C'_{Obs}(\theta)$ at WCS. Hereafter, $C_{Obs}$ and $C_{Sim}$ and their anomaly refer to
values at WCS. $C'_{Obs}(\theta)$ is defined:
$C'_{Obs}(\theta) = C_{Obs}(\theta)\text{-min}(C_{Obs}(\theta))$ (1)
with the minimum typically from the west. Specifically, $C'_{Obs}(\theta)$ was calculated by subtracting the minimum in the
annualized observed $C'_{Obs}(t,\theta)$ each year, $t$, after applying a 7-year running average.

Emissions from suburban communities, light industry, and commercial centers enhance $C'_{Obs}(\theta)$ for the north to east
(~350-70°) sectors. Removal of these terrestrial emissions was by fitting a Gaussian function to $C'_{Obs}(\theta)$ for
$330°<\theta<30°$ with the residual yielding $C'_{Obs}(\theta)$. This only affected $C'_{Obs}(\theta)$ for directions corresponding to the fields'
eastern edge.

Simulations were run at angular resolutions of 2°. Higher angular resolution produced small-scale artifacts for the
22/56-m sonar grid while the 11-m sonar grid was overly sparse due to the distance between sonar tracks (**Supp. Fig.**
**S3**).

The modeled source is from ω (in decibels), whereas emissions are moles m$^{-2}$ s$^{-2}$. Given that the relationship between
ω and bubble density (emissions) is complex and non-linear (Leifer et al., 2017), there is poor agreement between
$C'_{Sim}(\theta)$ and $C'_{Obs}(\theta)$. Thus, a correction function, $K(\theta)$, is applied to emissions for each grid, $E(i,j)$, along each $\theta$ and
the model rerun. $K(\theta)$ is:
$K(\theta) = {C'_{Obs}(\theta)}\Big/{C'_{Sim}(\theta)}.$ (2)
Initially, $K$=1, but in subsequent iterations, $K(\theta)$ is scaled as in **Eqn. 2** to adjust $E$ in cells along $\theta$. Because $K(\theta)$
weights closer seeps more than more distant seeps, a normalized distance varying correction function, $K(r,\theta)$, was
calculated such that,
$\int_{r=0}^{r=\infty} E(r,\theta) = \int_{r=0}^{r=\infty} K(r,\theta)\,E(r,\theta)\,dr$ (3)
where $r$ is distance from WCS. Simulations that shifted WCS northwards showed $E$ was varying nearly linearly.
Accounting for off-axis plume contributions requires several iterations to achieve *Convergence* defined,
$Convergence = {\sum C'_{Sim}(\theta)\,\sum C'_{Obs}(\theta)}\Big/{\sum C'_{Obs}(\theta)}$ (4)





Iterations were stopped after achieving *Convergence* of 1% or better – typically 4 to 5 iterations. Simulations suggested
wind veering, $\psi$, was important, which was implemented by calculating Gaussian plumes for $\theta$ and assigning it to
$C'(\theta + \psi)$.

**3 Results**
**3.1 Offshore *in situ* surveys**
An offshore COP seep field survey measured *in situ* $C_{CH4}$ and $u$ on 28 May 2016. Data were collected from the Santa
Barbara harbor (~7.5 km east of the seep field, **Fig. 2A; Supp. Fig. S6**) to offshore Naples, several kilometers west
of the seep field. Winds were fairly consistent easterlies over most of the seep field. Winds had an onshore component
near Campus Point and a broad (6-km wide) offshore flow west of COP that shifts to along coast near Naples (**Fig.**
**2A, white arrows**). Observed winds veered ~10° from east to the west sides of the seep field, roughly comparable to
the shift in coastline orientation.

Plumes are apparent downwind of major seeps, with the largest associated with the Trilogy Seep (**Fig 2B**). Strong
plumes also are evident downwind of the La Goleta Seep and Patch Seep. Notably, the Seep Tent Seep plume was
very weak. The Seep Tent Seep was the dominant seep area in the COP seep field from its appearance in June 1973
(Boles, Clark, Leifer, & Washburn, 2001) until recent years.

Additionally, the offshore survey identified focused plumes from beyond the extent of the seep field's 2005-sonar
map. Specifically in the Goleta Bay, which has been noted (Jordan et al., 2020), and offshore Haskell and Sands
Beaches, areas of abandoned oil wells.

Plume alkane $C'$ were determined by the difference between two "background" air samples collected immediately
outside the plume and three Trilogy Seep plume air samples. CH4 was 88.5% of THC, with ethane, propane, and
butane at 3.1%, 4.2%, and 2.76%, respectively, with pentane, hexane, and heptane at 1.11, 0.13, and 0.04%,
respectively (**Table 1**). THC molecular weight is 19.6 g mole$^{-1}$ based on a composition weighting. Branched alkanes
were detected, with 2-methylpentane and 3-methylpentane comprising 0.21%, each, as well as simple aromatics, e.g.,
benzene and toluene, with concentrations of 0.044 and 0.100 ppm, respectively.

The observed wet gas fraction ($C_2$-$C_5$/$C_1$-$C_5$) was 0.11 indicating a thermogenic origin (greater than 0.05 (Abrams,
2017)) and derivation from marine organic materials. Although the olefins ethene and ethyne were detectable at 0.02%
and 0.004%, respectively, butene was not detected. These olefins primarily derive from microbial processes (Abrams,
2017), thus, the ethane/ethyne ratio of 6200 also strongly indicates a thermogenic source (Bernard et al., 2001).
Atmospheric $CO_2$ was elevated by 12 ppm. Given that $CO_2$ completely dissolves from bubbles well before reaching
the sea surface (Clark et al., 2010), this demonstrates vertical transport of enhanced dissolved gases to the sea surface.






Plumes for the Trilogy Seeps, La Goleta Seep, and Seep Tent Seep were inverse modeled to derive emissions using
the average winds across each plume. For the Trilogy Seeps, $u$ was 5.9 m s$^{-1}$, insolation full sun, and the source height
was 25 m based on Trilogy's atmospheric plume being buoyant. Model surface concentrations for Trilogy B plume
are shown in **Fig 2A**. The other two seeps are far less intense and used a 1-m source height.

$E$ for Trilogy A was 1.28 Gg CH$_4$ yr$^{-1}$ (5600 m$^3$ CH$_4$ dy$^{-1}$), whereas Trilogy B and C contributed 0.06 and 0.07 Gg
CH$_4$ yr$^{-1}$, respectively, for a total of 6200 CH$_4$ m$^3$ dy$^{-1}$. Note, the plume origins and the sonar locations do not precisely
match because bubble surfacing location / atmospheric plume source are deflected by currents from the sonar-mapped
seabed vent location. Deflections can be up to ~40 m. La Goleta Seep released 4000 m$^3$ CH$_4$ dy$^{-1}$ and the Seep Tent
Seep released 310 m$^3$ CH$_4$ day$^{-1}$ with almost no surface bubble expression. For comparison, Clark et al. (2010) used a
flux buoy, which measures near surface bubble fluxes, and found Trilogy Seep emissions of 5500 and 4200 m$^3$ THC
dy$^{-1}$ and 930 m$^3$ THC day$^{-1}$ for La Goleta Seep in 2005 and 5700 m$^3$ THC dy$^{-1}$ for the Seep Tent Seep in 2002. During
the cruise, surface bubble plumes were not observed for the Seep Tent Seep, although its bubble plume had been a
perennial and dominant feature since its appearance. Note, Clark et al. (2010) reported THC in near sea surface bubbles
was 91% CH$_4$.

**3.2 West Campus Station**
**3.2.1 Temporal trends**
WCS is 500 m from the coast (to the southwest) at 11-m altitude and 850 m almost due south to the 11-m altitude
bluffs of Coal Oil Point (**Fig. 1**). Terrain slopes gently towards the coast to the southwest and towards a lagoon to the
south-southeast, rising again to the southeast to the COP bluffs. This flat relief likely has small to negligible effect on
wind speed and direction, although differential land-ocean heating could influence winds. Wind veering for the coast
to the east of COP is likely due to the orientation of the coastline and bluffs.

The WCS improvements in 2008 (**Fig. 3-dashed line**) allowed far higher values of $C$ and $u$ (**Supp. Fig. S7A,7B**).
Comparison of the probability distributions of $u$ and $C$, $\phi(u)$ and $\phi(C)$, respectively, before and after the upgrade did
not suggest biases were introduced (**Supp. Fig. S7C,7D**). Specifically, changes in the average and median values and
in the baseline after 2008 were from better measurement of higher value events (gusts and short positive $C$ anomalies).

Significant daily to seasonal to interannual variations are apparent in the daily-averaged $u$ and $C$ (**Fig. 3**). The calmest
season is late summer to fall, whereas spring is the windiest with greatest variability due to synoptic systems (**Fig.**
**3A**). Winds have strengthened since a minimum in 1995-1996, moreso for the seep directions with stronger winds
becoming more frequent, moreso summer than winter (**Supp. Figs. S8, S9**).





Trends in $C$ reflect trends in both seep field emission and ambient $C$. $C$ is higher in fall and spring (**Fig. 3B**). Given
that stronger winds decrease $C$ through dilution, this suggests the seasonal variation in $C$ underestimates the seasonal
variation in emissions. Several studies have shown increased emissions under higher wave regimes (storminess),
reviewed in Leifer (2019) and proposed from wave pumping. Storms increase evasion from higher wave turbulence
and breaking-wave bubbles, which sparge dissolved $CH_4$ and other trace gases down to the seabed in shallow (<100
m) waters (Shakhova, Semiletov, Salyuk, et al., 2010). Note, $u$, $\theta$, and $C'$ correlate with time of day. For example,
north generally reflects weak, offshore nocturnal winds with no seep contribution.

**3.2.2 Spatial heterogeneity**
Calculating the angular-resolved average $C$, $C_{ave}(\theta)$, for the complete dataset with respect to $\theta$ shows the highest $C$
from the main seep field direction (155-250°, **Fig. 4**). For the seep directions, $C_{ave}(\theta)$ was poorly fit by a single
Gaussian function but well fit ($R^2$=0.997) by two Gaussian functions with peaks at 178° and 198° corresponding to
the Seep Tent and Trilogy Seeps' directions, respectively (**Fig. 4A, 4B**). Notably, the fit residual showed a linear
increasing trend, $dC_{ave}(\theta)/d\theta$, of 0.17 ppb degree$^{-1}$ from 180 to 210° (**Supp. Fig. S9B**) consistent with evasion from
a dissolved downcurrent plume that drifts west-northwest along the coast (Leifer, 2019).

The average $C$ anomaly, $C'_{ave}(\theta)$, was calculated from the average of $C_{Obs}(\theta)$ after **Eqn. 1** with terrestrial
anthropogenic sources to the north to northeast were removed. The minimum in $C_{Obs}(\theta)$ was at 270°, a direction with
no mapped seepage at the dissolved plume's approximate shoreward edge.

There is a strong, focused peak in $C_{max}(\theta)$ at $\theta$~190°, close to the Seep Tent Seep direction (**Fig. 4E, 4F**), which is
fairly isolated on the Ellwood Trend (**Fig. 1**). This peak also is close to the direction of Tonya Seep on the Inshore
Trend and to the small, unnamed area of seepage to the west of Trilogy Seep along the Red Mtn. Fault trend. The $\theta$-
resolved maximum $C(\theta)$, $C_{max}(\theta)$, remains elevated through ~270°, far west of the $C_{ave}(\theta)$ peak at ~200°. This
strongly suggests that the seep field extends further to the west-northwest than current maps. These data cannot be
explained by dissolved plume outgassing, which would affect $C_{ave}(\theta)$ but not $C_{max}(\theta)$.

$C(\theta)$ enhancements for non-seep directions (**Fig. 4A,4B**) show a peak at ~35°, corresponding to the direction of a
commercial center amid suburban development. This could result from terrestrial seepage and natural gas pipeline
leakage and/or THC emissions from communities and traffic.

Neglecting the synoptic system, topographic forcing from the east-west Santa Ynez range means that prevailing winds
are westerlies, which also are the strongest (**Fig. 4C, 4D**). North winds (320-15°) largely are weak as are winds from
due south; however, the sea breeze strengthens winds rapidly away from due south. $\theta$ peaks in the maximum winds
(1-minute sustained), $u_{max}(\theta)$, correspond to the west and east peaks in $u_{ave}(\theta)$ with strengths to 16 m s$^{-1}$. Interestingly,





there also are strong north (0-30°) winds or downslope flow, termed sundowner winds, a highly localized and
infrequent phenomenon. The overlap of $u_{med}(\theta)$ and $u_{ave}(\theta)$ shows winds largely are normally distributed.

The wind speed probability distribution, $\phi(\theta,u)$, defined:
$$\int^u \phi(\theta,u)du = 1 \qquad\qquad (5)$$
varies significantly with direction (**Fig. 5A**). The distribution is very narrow (y-axis) for the northeast (~45°) where
winds are largely weak. The distribution is broad for the east-southeast (70-135°) and for the prevailing westerlies
(250-280°). The east-southeast distribution skews to the south (stronger winds extend further from the south -
offshore), whereas the prevailing westerly wind distribution skews to the northeast (as does the coastline).

The median $C$, $C_{med}(\theta)$, and average $C$, $C_{ave}(\theta)$, have similar shapes, albeit with lower values at all $\theta$ (**Fig. 4A**),
indicating $C$ is not normally distributed. This is shown in the $\theta$-resolved probability distribution of $C$, $\phi(C,\theta)$ (**Fig.**
**5B**). In the seep direction, $\phi(C,\theta)$ extends to much higher values than from non-seep directions. $\phi(C,\theta)$ is asymmetric
with $\theta$ extending further to the west than the seep field extent (240°) and then decreasing more abruptly than the
decrease to the east. This asymmetry is expected given the seep field's asymmetric orientation relative to WCS (eastern
seepage is more distant). Emissions beyond the field's mapped western edge arise from downcurrent plume outgassing
and potentially a contribution from unmapped seeps.

**3.2.3 Seep field diurnal emissions cycle**
$C$ and $u$ for the seep field direction, $u_{seep}$, and $C_{seep}$, respectively, follow diurnal patterns that are not the same as the
overall diurnal pattern due to the wind direction constraint and because $C_{seep}$ depends on $u_{seep}$. The dependency arises
because higher $u$ dilutes emissions, decreasing $C$, but higher $u$ also increases dissolved plume evasion and bubble-
mediated emissions from higher swell (after a delay for wave buildup). Diurnal winds in coastal regions feature a shift
between weak nocturnal offshore winds that veer to onshore winds in the morning - the sea breeze circulation. This
was explored in time and direction segregated $u$ and $C$ and seep direction averaged $u_{seep}$, and $C_{seep}$ for 90-270° (**Fig.**
**6**). Data were segregated by $\theta$ for pre- and post-2008 (when station improvements facilitated better wind
characterization, particularly night time, which are seldom from the seep field direction, see **Supp. Fig. S10** for 1991-
2007). $u(\theta,t)$ and $C(\theta,t)$ were 2D Gaussian kernel smoothed with a 1-bin standard deviation (contours based on a 3 bin
standard deviation) by the imgaussfilt.m algorithm (MATLAB, MathWorks, MA) after interpolating the calibration
data gap 24:00-01:00.

Early morning (01:00–03:00) $u_{seep}$ are stronger because typical nocturnal winds are northerlies (land breeze), coming
from the south largely during storms. These are accompanied by elevated $C_{seep}$ implying greater emissions despite
enhanced dilution from stronger winds. The minimum in both $u_{seep}$ and $C_{seep}$ occur in the early morning (04:00-08:00),



with both increasing slightly through midday (~12:00). $C_{seep}$ follows an afternoon trend of overall decreasing to a
minimum at ~ 20:00 before increasing again into the late evening.

Underlying these trends are complex temporal spatial patterns. $u$ for the north to northeast reaches a maximum around
noon and peak around 16:00; while $C$ for northeast to east is low in the morning reaching a peak to the east in the
afternoon and likely reflects terrestrial sources. This pattern in $C(t,\theta)$ extends to nearly 130°. Beyond the seep field's
western edge, $u$ is elevated from the prevailing direction (270°), with $C$ elevated throughout the morning. There also
is a short-lived peak in $u$ around noon at ~300°, which corresponds to a short-lived depressed $C$. These could be
consistent with wave development time, transport time, and sparging of the downcurrent plume; however,
interpretation based on these spatial patterns largely is speculative.

**3.3 Overall seep field emissions**
**3.3.1 Overall emissions**
Average atmospheric emissions, $E_A$, for 1990-2020 were derived by an iterative Gaussian plume model, initialized
with the 2005 sonar map (**Fig. 1**). An emissions sensitivity study on the effect of grid resolution was conducted for
resolutions from 11 to 225 m and a 22/56-m hybrid grid (**Fig. S3**). Simulations used moderate insolation to derive the
turbulence parameters and stability class, a 250-m $BL$ (typical Santa Barbara Channel marine values (Edinger, 1959;
Rahn, Parish, & Leon, 2017)), and 2° angular resolution (Hanna et al., 1982). Simulations were run iteratively until
convergence, typically within 5 iterations (**Supp. Fig. S11**). Simulations used a linear distance weighting function,
$K(r,\theta)$, based on sensitivity study (**Supp. Fig. S12**).

Simulations could not reproduce observations in the Platform Holly direction ($\theta$=238°). Thus, a source was added for
the platform area, which improved simulation-observational agreement in this wind direction. Since significant seep
bubbles plumes generally are not observed in the platform's vicinity, these emissions could arise from incomplete
combustion from flaring.

The model-derived, $E_A$, for 1990-2020 was 83,500 m$^3$ dy$^{-1}$ (**Fig. 7**). Using a composition-weighted molecular mass of
19.6 g mole$^{-1}$ implies 27 Gg yr$^{-1}$. Atmospheric seep gas is 88.5% $CH_4$, implying 19 Gg $CH_4$ yr$^{-1}$ seep emissions (**Table**
**1**). Given that $CH_4$ is 73% of THC, non-methane hydrocarbon (NMHC: $C_2$-$C_7$) emissions are 9,500 m$^3$ dy$^{-1}$. The largest
NMHC was propane with emissions of 3510 m$^3$ dy$^{-1}$, followed by ethane at 2590 m$^3$ dy$^{-1}$.

Seabed emissions, $E_B$, are necessarily significantly greater than $E_A$ in the near field as $E_A$ misses the fraction of
emissions that remain in the water column, $E_W$. Note, some of the $E_W$ fraction in the near field evades to the atmosphere
in the far field. Clark et al. (2000) estimated a 50:50 air/water partitioning, implying seabed emissions, $E_B$, 1990-2020
of 167,000 m$^3$ dy$^{-1}$ or 54 Gg yr$^{-1}$. A comparison of $E_A$ versus $\omega$ showed a very steep increase with $\omega$ for $E_A$ = 1-10 g s$^{-1}$
$^{-1}$ m$^{-2}$ with rollover at $\omega$~0.025 (**Supp. Fig. S13**).






Insights into the simulations were provided by how the model partitioned emissions between different seep areas.
Particularly notable is the model's treatment of the Trilogy Seep area - the second strongest seep area after the Seep
Tent Seep - through much of the study period. The model re-assigned Trilogy Seep emissions to seepage to the west,
representing Trilogy Seep emissions as unrealistically weaker than other, smaller seeps, such as IV Super Seep. One
contributor to this re-assignment is the diurnal cycle. Specifically, morning winds are weaker and from the south and
east while afternoon winds are stronger and from the west, thus weaker winds are used to "measure" emissions from
the eastern field and stronger winds are used to "measure" emissions from the western field. This could bias emissions
(higher for the west and lower for the east) due to an emissions dependency on $u$ – note, wind dilution is addressed by
the Gaussian Plume model.

The model assigned strong emissions to the field's eastern and western edges even though sonar returns here are small.
This suggests wind veering plays an important role at the seep area scale. In a comparison of the Seep Tent Seep and
La Goleta Seep areas, the model emphasized the Seep Tent Seep whereas La Goleta Seep emissions were shifted to
inshore seepage. This re-partitioning was greatly reduced for a +10° wind veer, which also lessened the strengthening
of emissions from the field's western edge relative to sonar (**Supp. Fig. S14**). Given the lack of field data between the
seep field and WCS on wind veering, further wind veering analysis was not conducted.

**3.3.2 Seep field sector emissions**
To investigate sub-field scale emissions, the seep field was segregated into three sectors: inshore, offshore east, and
offshore west (**Fig. 1**). Based on integrating sonar return, $\omega$, the inshore seepage contributes 40% of the field's $\omega$ with
the offshore seep trend split between 9% for the west and 51% for the east. Supporting this comparison is the similarity
in the normalized sonar return probability, $\phi_n(\omega)$, for the inshore seeps and offshore east seeps (**Fig. 8**). In contrast,
$\phi_n(\omega)$ for the offshore west seepage differed dramatically despite the similarity in geology along the anticline
underlying the offshore seep trend (Leifer et al., 2010). This likely results in part from the interaction between
migration and production. Although the normalized atmospheric emissions probability, $\phi_n(E_A)$ for the inshore and
offshore seeps are similar over most of the range (except the weakest, $E_A < 0.02$ g s$^{-1}$), significant differences are evident
between offshore east and west seepage. Offshore east seepage is more dispersed and favors weaker seepage compared
to offshore west seepage and compared to $\phi_n(\omega)$.

The weakest seepage ($\omega < 0.02$) contributes negligibly to overall emissions and had no notable inshore-offshore
difference in $\phi_n(\omega)$. The largest difference is between the strongest seepage ($\omega > 0.5$) for the inshore and offshore seeps.
Specifically, there is a strong peak at $\omega \sim 0.45$ and nothing stronger for the inshore seeps, whereas offshore $\phi_n(\omega)$
continued to larger $\omega \sim 0.7$. The inversion reduced $\phi_n(E_A)$ for the strongest inshore compared to $\phi_n(E_A)$ for strong
offshore seepage. Also, $\phi_n(E_A)$ was reduced far more for offshore east seepage than offshore west seepage.



These distributions suggest that controlling geological structures (fractures, fault damage zones, and chimneys in the
capping Sisquoc Formation) are the same for inshore seepage and offshore east seepage, with the primary difference
for the strongest seepage in these two sectors which are of similar strength– the inshore Trilogy Seeps provide focused
emissions, whereas the offshore east La Goleta Seeps are comparatively dispersed and far oilier.

Although, $\omega$ is not emissions, modeled $E_A$ followed the 40:60 partition in $\omega$ between inshore and offshore seepage.
Interestingly, the $E_A$ partitioning between the offshore east versus the offshore west differed significantly from
partitioning for sonar with 21% from offshore west and 38% from offshore east, i.e., this greatly accentuated the $E_A$
Seep Tent Seep area. In part, this arises from a diurnal cycle bias – WCS observes the offshore west seeps for
afternoon/evening westerly winds, which are stronger, whereas WCS observes the offshore east seeps when winds are
weaker, earlier in the day (**Fig. 6B**). Winds increase bubble emissions from wave hydrostatic pumping and dissolved
gas evasion. Also potentially contributing is saturation of $\omega$ at very high bubble-density bubble plumes, such as the
Seep Tent Seep and Trilogy Seep (Leifer et al., 2017). Saturation would imply an under-estimate of $\omega$ for the strongest
seep area emissions.

**3.3.3 Uncertainty and emissions sensitivity**
Given the number of sources of variability that are poorly characterized by available data, uncertainty would be best
assessed by Monte Carlo simulations; however, this was unfeasible due to the simulations' computational demands.
Thus, emissions uncertainty was investigated by sensitivity studies (**Fig. 9**). Where data were available, uncertainty
due to a specific parameter was estimated from these simulations. Specific factors studied included sonar resolution,
angular resolution, $\delta\theta$, wind speed, $u$, concentration anomaly, $C'$, boundary layer height, $BL$, wind veering, $\psi$, spatial
northing offset, $Y$, and the inshore and offshore seepage partitioning, $\zeta$. Sensitivity studies are detailed in **Supp. Sec.**
**S7.4.**

The contribution to uncertainty from $\delta\theta$, $C'$, $\psi$, and spatial offsets within the seep trends were minimal – just a few
percent or less, as was uncertainty associated with $u$ due to $BL$ sensitivity countering $u$ sensitivity. Moderate
uncertainty was identified for $BL$ and $\zeta$. For $BL$ ranging from 150 to 350 m, mean $E_A$ uncertainty was 6%. Assessing
uncertainty in $\zeta$ was more challenging as there is no verification data on variability in the $E_A$ partitioning between the
inshore and offshore seep trends. Still, the mean $E_A$ uncertainty for $-50\%<\zeta<50\%$ is 11.5% from a polynomial fit.
Whereas $\zeta$ could be larger, there is consistency in seepage location between sonar surveys spanning decades (Leifer,
2019), which suggests that averaged changes in $\zeta$ on multi-decade timescales are modest. Total uncertainty was taken
as 15% based on the sum of uncertainty in $BL$ and $\zeta$.





**3.4 Ellwood Field emissions**
$C(\theta)$ increases to the northeast with a peak at 290-320° corresponding to the direction towards abandoned wells off
Haskell Beach (**Fig. 10**). Emissions from this area – either from seepage or leaking wells also were noted in the
offshore survey data near Haskell Beach (**Fig. 2A**). Additionally, $C_{max}(\theta)$ shows a 22-ppm peak in in this direction
(**Fig. 4F**), also consistent with natural seepage or well leakage.

Ellwood field production continued through the 1970s with wells drilled into the geological structures that allowed
oil accumulation (Olson, 1983) but also provided migration pathways (Leifer et al., 2010). There are many abandoned
wells from these fields and others fields in the Goleta Plains, beaches, and shallow near-coastal water to the west-
northwest of WCS (offshore Haskell Beach and onshore around Naples Point). Currently, active wells only are found
at the La Goleta Gas field (a natural gas storage field), east of WCS.

Faults associated with these anticlines provide migration pathways and are aligned approximately with the coast in a
series of roughly parallel faults extending onshore (Minor et al., 2009). The onshore/coastal Ellwood field (northwest
of the South Ellwood field) sources from the primarily sandstone Vaqueros Formation (Olson, 1983), whose main
trap is an anticline at the western edge of the North Branch Western More Ranch Fault (NBWMRF). Offshore seepage
tracks some of these faults, e.g., the Isla Vista Fault trend corresponds to an offshore seep trend in Goleta Bay that
includes the Goleta Pier Seep, whereas wells follow the NBWMRF trend offshore of Haskell Beach.

**4    Discussion**
**4.1 Atmospheric seep field observations**
**4.1.1. Air quality station**
A range of approaches are available to evaluate marine seepage $CH_4$ emissions: *in situ* approaches including direct
capture (Leifer, 2015; Washburn, Johnson, Gotschalk, & Egland, 2001), fluid flow measurements (Leifer & Boles,
2005), video (Leifer, 2015), and remote sensing approaches that include active acoustics, i.e., sonar (Hornafius et al.,
1999), dissolved *in situ* (Marinaro et al., 2006), and passive acoustics (Wiggins et al., 2015). Remote sensing is the
best approach for long-term monitoring to capture emissions shifts between vents. To date, only sonar remote sensing
has provided quantitative seep plume (seabed) emissions. Notably, sonar ranges are up to a few hundred meters, far
less than the size scales of many seep fields, while high power-demands typically require a cabled observatory for
long-term observations.

This study demonstrated that air quality station data provides the long-term continuous data needed to capture seasonal
variations including emissions during storms and transient events, which field campaigns likely miss. For example,





sonar surveys tend to occur during summer when seas are calmer and more predictable but when seepage is weakest
(**Fig. 3**), remaining in port during storms when emissions are enhanced.

The approach derived atmospheric trace gas emissions for a dispersed area source constrained by sonar seepage maps
from long-term air quality and meteorology data. This approach can be extended to terrestrial seepage if the source
can be constrained spatially (due to geology); although nearby anthropogenic sources may complicate emissions
assessment. Other terrestrial sources such as landfills, O&G production fields, or industrial sites – if spatially
constrained – could be addressed by this approach. The use of cavity enhanced absorption spectrometers that can
speciate gases like $CH_4$ and $C_2H_6$ could enable discrimination of confounding sources as well as better characterization
of emissions. Although an onshore station can address nearshore seepage, further offshore seepage could be addressed
by a moored station. A moored station could also support *in situ* aqueous chemical sensors, current measurements.

**4.1.2 *In situ* atmospheric surface surveys**
Atmospheric emissions were assessed for three seep areas by an atmospheric *in situ* survey approach wherein
downwind data are collected orthogonal to the wind direction in a transect that spans the plume (background to
background on the plume's edges). This approach was developed for terrestrial sources (Leifer, Melton, Tratt, et al.,
2018) yet largely has not been used for offshore marine seepage, which often are area sources. In this study, this was
addressed by gridding the area source and treating each grid as a far field point source. Gaussian plume inversion
requires distant source(s), i.e., far field. Surveys of three strong seep areas all were well characterized by the Gaussian
plume model.

One advantage of atmospheric surveys is rapidity - a single transect of a few minutes is sufficient to derive emissions
for a seep area. In comparison, a flux buoy survey can require many hours to a day (Clark et al., 2010), during which
forcing factors (waves, tides, etc.) change significantly. Also rapid are seep area sonar surveys (Wilson, Leifer, &
Maillard, 2015) allowing a combined sonar and atmospheric survey to repeat characterize emissions and sea-air
partitioning within a few hours. With respect to the entire COP seep field, whereas a sonar survey requires two to
three days (Leifer et al., 2010), a downwind atmospheric survey is far more rapid, requiring perhaps an hour. This
allows repeat field emissions measurements over a tidal cycle.

**4.2 Seep field emissions**
**4.2.1 Total emissions**
To date, only two estimates of COP seep field seabed emissions, $E_B$, have been published. Hornafius et al. (1999)
estimated $E_B=1.5 \times 10^5$ m$^3$ dy$^{-1}$ (64 Gg yr$^{-1}$) based on sonar surveys covering 18 km$^2$ from Nov. 1994 – Sep. 1996,
collected during the summer to late fall seasons. This value excluded Seep Tent collection. A 4.1 km$^2$ sonar survey in
Aug.-Sep. 2016 estimated $E_B=24,000$ m$^3$ dy$^{-1}$ (Padilla, Loranger, Kinnaman, Valentine, & Weber, 2019), significantly



lower, which in part arises from field subsampling, but also could arise from long-term changes; however, neither
study addressed temporal variability. The sonar surveys occurred in summer and fall when seepage activity is at a
minimum, whereas winter and early spring feature much higher activity associated with large transient events and
storms (Bradley et al., 2010).

Hornafius et al. (1999) used an engineered bubble plume to calibrate emissions, an approach also used in Leifer et al.
(2017). Due to technology limitations at the time, the strongest seepage was clipped or saturated, i.e., underestimated,
and the survey did not cover shallow seepage. Thus, the Hornafius et al. (1999) emissions estimate is a lower limit for
summer/fall emissions. The Padilla et al. (2019) survey was calibrated by an inverted seep flux buoy suspended at 23
m. This differs significantly from seep flux buoy measurements (Washburn et al., 2001), which are collected in surface
drift mode. Surface drift mode ensures a horizontal orientation for the buoy and an absence of lateral velocity
difference between the capture device and currents – either of which decreases capture efficiency from 100%, biasing
derived emissions low. Further, the Padilla et al. (2019) survey was calibrated 1 month after the sonar surveys, whereas
the 1995 engineered plume calibration by Hornafius et al. (1999) was contemporaneous. The Hornafius et al. (1999)
approach accounts (partially) for dissolution between the seabed and survey depth window, albeit air dissolves slower
than methane. Dissolution losses between the seabed and the depth window can be addressed by a numerical bubble
model (Leifer et al., 2017).

Based on the Gaussian plume model-derived $E_A$ was $8.4 \times 10^4$ m$^3$ dy$^{-1}$, which based on a Clark et al. (2000) assessment
that half the seabed seepage reaches the atmosphere, suggests $E_B = 1.7 \times 10^5$ m$^3$ dy$^{-1}$; very similar to $E_B = 1.5 \times 10^5$ m$^3$ dy$^{-1}$
$^{-1}$ from Hornafius et al. (1999). This agreement is coincidental as it neglects seasonal and interannual trends. For
example, Bradley et al. (2010) found 1994-1996 emissions were well below the average for 1990-2008, increasing
significantly after 2008.

**4.2.2 Methane and non-methane hydrocarbon emissions**

Analysis of atmospheric samples provided a picture of the complexity of atmospheric emissions that arises from the
multiple pathways underlying atmospheric emissions. Specifically, as bubbles rise, they lose lighter and more soluble
gases faster (deeper in the water column), leading to differences between evasion from dissolved gases and direct
bubble transport (Leifer & Clark, 2002). Thus, bubble-mediated transport enhances larger alkanes relative to smaller
alkanes leaving more of the smaller alkanes in the water column. For strong seeps, bubble plumes are associated with
strong upwelling flows (Leifer et al., 2009), which transport dissolved gases to the sea surface where they can outgas.
Additionally, oil (as droplets and bubble coatings) enhances alkane transport due to slower dissolution and diffusion
of larger alkanes through oil.

Atmospheric plume concentrations were 11.5% NMHC and 88.5% CH$_4$, very similar to Hornafius et al. (1999) who
referenced the Seep Tent composition (88% CH$_4$, 10% NMHC, and 2% nitrogen) as very similar to the reservoir





composition. Note, Clark et al. (2010) observed Trilogy near sea surface bubbles with 5.7% to 7.9% NMHC and 52.4
to 79.7% $CH_4$, demonstrating significant partitioning. The similarity between the atmospheric and seabed composition
demonstrates efficient dissolved gas transfer to the sea surface.

COP seep field seabed emissions are orders of magnitude greater than typically reported for other seep areas, e.g.,
summary Römer et al. (2017) where emissions for 12 different seep areas including for sites in the North Sea, Pacific
north west, Gulf of Mexico, etc., were 2-480 tons $yr^{-1}$, multiple orders of magnitude less than seabed emissions for
Coal Oil Point. Römer et al. (2017) for Dogger Bank in the North Sea observed atmospheric $CH_4$ plumes and estimated
direct atmospheric bubble-mediated transfer using a bubble model to suggest 20% of seabed emissions directly
reached the atmosphere 21.7 ton $yr^{-1}$. For the Tommelieten Seeps in 70-m water Schneider von Deimling et al. (2011)
estimated 4% of the 0.024 Gg $CH_4$ $yr^{-1}$ seabed emissions, i.e., ~1 Mg $CH_4$ $yr^{-1}$ reached the atmosphere by bubble-
mediated transfer. Schneider von Deimling et al. (2011) used a bubble model based on an assumed bubble size and
neglected diffusive flux. These diffusive fluxes include bubble dissolution into the wave mixed layer in the local area.
A few studies have directly measured atmospheric fluxes by an air-sea gas transfer model. For example, Schmale,
Beaubien, Rehder, Greinert, and Lonmbardi (2010) found seep air fluxes of 0.96-2.32 nmol $m^{-2}$ $s^{-1}$, much higher than
the ambient Black Sea flux of 0.32-0.77 nmol $m^{-2}$ $s^{-1}$. In the Black Sea, ambient emissions arise from microbially
produced $CH_4$ in shelf and slope sediments (Reeburgh et al., 1991). Di, Feng, and Chen (2019) estimated 7.7 nmol $m^{-2}$
$s^{-1}$ for the shallow South China Sea based on an air-sea gas transfer model. If we disperse COP seep field atmospheric
emissions of $1.15 \times 10^9$ M $yr^{-1}$ over the ~6.3 $km^2$ of 25x25 $m^2$ bins with emissions, we find 5.7 $\mu M$ $m^{-2}$ $s^{-1}$, three orders
of magnitude greater.

Recent estimates of total global geo-$CH_4$ sources from a bottom-up approach are 45 Tg $yr^{-1}$ with submarine seepage
contributing 7 Tg $yr^{-1}$ (Etiope & Schwietzke, 2019), implying COP seep field contributes 0.25% of the bottom-up
submarine emissions. However, an estimate of pre-industrial $CH_4$ emissions (not confounded with fossil fuel
production emissions) based on ice core $^{14}CH_4$ suggested 1.6 Tg geo-$CH_4$ $yr^{-1}$ emissions (Hmiel et al., 2020). This
estimate, if accurate, suggest the COP seep field contributes an astounding 1% of global seep emissions (submarine
and aerial) and is difficult to reconcile with the COP seep field and other top seepage estimates. For example, $CH_4$
atmospheric emissions for the Lusi hydrothermal system of 0.1 Tg $yr^{-1}$ (Mazzini et al., 2021), a hotspot in the Laptev
Sea of 0.9 Tg $yr^{-1}$ into shallow seas (Shakhova, Semiletov, Leifer, et al., 2010), and for the East Siberian Arctic Sea
using eddy covariance of 3.0 Tg $yr^{-1}$(Thornton et al., 2020). Thus, COP seep field emissions either play a significant
role in global seep emissions or indicate that geo-gas emissions are less tightly constrained.

COP seep field $C_2H_6$ emissions were 1.27 Gg $C_2H_6$ $yr^{-1}$. For reference, this is 11% of the 11.4 Gg $C_2H_6$ $yr^{-1}$ in 2010
for the South Coast Air Basin (SCAB), which includes Los Angeles (Peischl et al., 2013). Globally, Simpson et al.
(2012) and Höglund-Isaksson (2017) found 11.3 and 9.7 Tg $C_2H_6$ $yr^{-1}$ in 2010, respectively. $C_2H_6$ has been increasing
since 2010 due to increased O&G production emissions (Helmig et al., 2016). Globally, seeps are estimated to





contribute 2-4 Tg $C_2H_6$ yr$^{-1}$ (Etiope & Ciccioli, 2009), and from ice cores, 2.2-3.5 Tg yr$^{-1}$(Nicewonger, Verhulst,
Aydin, & Saltzman, 2016), suggesting the seep field contributes 0.03-0.06% of global seep emissions.

Seep THC was 4.2% propane, implying emissions of 2.5 Gg $C_3H_8$ yr$^{-1}$. Global propane emissions are 10.5 Tg yr$^{-1}$
(Pozzer et al., 2010), with 1-2 Tg yr$^{-1}$ estimated for seeps (Etiope & Ciccioli, 2009). Thus, the COP seep field could
contribute 0.05-0.1% of the global seep budget. Oceans are estimated to contribute 0.35 Tg $C_3H_8$ yr$^{-1}$ (Pozzer et al.,
2010), less than geological seepage.

Global butane emissions are 14 Tg $C_4H_{10}$ yr$^{-1}$ (Pozzer et al., 2010), higher than ethane and propane. COP seep field
butane ($C_4$) and pentane ($C_5$) emissions were 2.2 Gg $C_4H_{10}$ yr$^{-1}$ and 1.1 Gg $C_5H_{12}$ yr$^{-1}$, respectively, thus combined $C_2$-
$C_5$ emissions are 7.1 Gg yr$^{-1}$, compared to 65 Gg yr$^{-1}$ from the entire SCAB, i.e., COP seep field contributes ~5% the
SCAB. COP $C_2$-$C_5$ emissions are significantly above that of the La Brea area, estimated at 1.7 Gg yr$^{-1}$ (D. Weber et
al., 2017). Note, COP seep field atmospheric $C_2$-$C_5$ emissions certainly are larger, potentially significantly, as higher
alkanes also are emitted from oil slicks, whose downcurrent contributions were not considered for this study and
atmospheric plume was not sampled for this study.

Both benzene and toluene were detected with estimated emissions of 5000 and 1300 kg yr$^{-1}$, respectively. These
emissions likely are underestimates, potentially significantly, due to neglecting the contribution from oil slicks. Both
gases are of significant health concerns.

Based on evaluating the COP seep field with respect to global seep ethane and propane emissions, COP seep field
contribution to global geo-$CH_4$ emissions (in parentheses) are consistent with recent global geo-gas $CH_4$ emissions
estimates of 45 Tg yr$^{-1}$ (0.04%) (Etiope et al., 2019), but not the significantly lower pre-industrial estimates of global
geo-$CH_4$ emissions, e.g., 1.6 Tg yr$^{-1}$ (1.15%) (Hmiel et al., 2020).

**4.3 Downcurrent emissions**
The seep field concentration, $C'(\theta)$, anomaly was centered at $\theta$~200° and was well described by a dual Gaussian
function (**Fig. 4B**). This was surprising given that the seep field is asymmetric with respect to a 200° axial line from
WCS to COP. Underlying this seeming discrepancy is that WCS winds are weakest from due south and strongest from
the west (prevailing) and also stronger to the east-southeast (**Fig. 4C**).

The residual of the Gaussian fit increased in the downcurrent direction (**Supp. Fig. S9B**), consistent with evasion from
the downcurrent dissolved plume plus and seepage from this area. The dissolved plume roughly follows the coast,
extending as far as ~280° from WCS due to the coastline shift from northwest to west around Haskell Beach (**Fig. 2**),
~30° beyond the seep field's sonar mapped western edge (**Fig. 1**). As prevailing winds are westerlies (paralleling the
coastal mountains), downcurrent plume evasions decrease with distance as surface waters become depleted by evasion.





Evasion increases non-linearly with $u$, particularly for winds that include wave breaking (Nightingale et al., 2000);
however, higher winds also dilute emissions more. Note, there are no mapped seeps in this area.

A similar situation likely occurs towards the east and leads the model to emphasize seepage at the field's eastern
extent, too. Specifically, strong prevailing afternoon westerly surface winds drive a near-surface dissolved plume
eastwards. When these westerly winds calm down late in the evening, easterly winds will transport evasion from this
east-displaced dissolved plume towards WCS. Additionally, it also is possible that the COP seep field extends further
east than mapped in sonar surveys, at least during some seasons.

**4.4 Focused seep area emissions**
Trilogy Seep area emissions were estimated at 6,200 m$^3$ CH$_4$ dy$^{-1}$ in May 2016. For comparison, Clark et al. (2010)
found 5500 and 4200 m$^3$ THC dy$^{-1}$ (4,900 and 3,700 m$^3$ CH$_4$ dy$^{-1}$) for Trilogy Seep as measured by flux buoy for near
surface bubble fluxes in Sept. 2005. Note, the plume inversion approach also includes outgassing of near surface
waters that have enhanced $C_{CH4}$ from plume dissolution, the flux buoy approach does not. Although Clark et al. (2010)
found surface bubbles had undetectable CO$_2$, the atmospheric plume's CO$_2$ to CH$_4$ concentration ratio was comparable
to the seabed bubble concentration ratio. This demonstrates significant upwelling flow transport of seabed water to
the sea surface where dissolved gases evade near where the bubble plume surfaces. This near-plume evasion
contributes to the atmospheric plume surveyed by the *in situ* measurements. Note, these emissions neglect
downcurrent emissions. A 50:50 atmosphere/ocean partitioning suggests 2016 Trilogy Seep emissions were ~40%
lower than in 2005 – a difference within the difference between the two 2005 Trilogy Seep measurements Clark et al.

753 (2010).


In contrast, there was very poor agreement for the Seep Tent Seep, for which Clark et al. (2010) mapped emissions of
5700 m$^3$ day$^{-1}$ (5000 m$^3$ CH$_4$ day$^{-1}$) in Nov. 2002 whereas this study found 310 m$^3$ CH$_4$ dy$^{-1}$. This discrepancy was
readily apparent with almost no visible surface bubble expression in May 2016, whereas the Seep Tent Seep has been
a perennial feature since its appearance. The absence of more than a few scattered bubbles at the sea surface (the boil
in 2000 was driven by a 1-2 m s$^{-1}$ upwelling - Leifer, Clark, and Chen (2000)) indicates that most emissions are from
evasion. A buoyancy plume associated with the rising oil (thick oil slicks surface above the Seep Tents) as well as
methane dissolved in the oil likely is transporting the observed, focused CH$_4$ emissions.

This is remarkable given that the seep field's geofluid migration "center" in recent decades has been the Seep Tent
Seep (Bradley et al., 2010), which was the largest seep in the field in 2010 (Clark et al., 2010). The Seep Tent Seep
consists of emissions not captured by the Seep Tents – two large (33 m square) steel capture tents on the seafloor. For
reference, the Seep Tents captured ~16,800 m$^3$ gas dy$^{-1}$ in the early 2000s (Boles et al., 2001). Bradley et al. (2010)
found in WCS data that when overall seep field emissions decreased to a minimum in 1995, they were focused on the





Seep Tent Seep direction. Note, the Seep Tent Seep was observed first in 1970 as a boil visible from 1.6-km distant.
The seepage was tented in Sept. 1982 (Boles et al., 2001).

Underlying these observations are several factors. First, the Seep Tent Seep is modern – since 1978 – as it was not
mapped in a 1953 seep survey (Leifer, 2019). At the time it was first reported as a sea boil visible over a kilometer
distant (Boles et al., 2001). Since installation, overall Seep Tent production has diminished (Boles et al., 2001) by a
factor of 3 from 1984 to 1995. Some fraction of this trend could have resulted from the expansion of active seepage
beyond the seep tents. Perhaps more significantly, the Seep Tent Seep lies over one of the Platform Holly wells (Leifer
et al., 2010; Fig. 3C), creating the potential of linkage between well production (including stimulation) and Seep Tent
production and thus Seep Tent seepage (the uncaptured portion). The near cessation of Seep Tent Seep emissions in
the field observations is consistent with a positive relationship between the two.

**4.6 Diurnal trend and bias**
The diurnal wind patterns typical of the coastal Pacific marine environment are weak offshore (northerly) night winds
that shift to from the east in the morning and then from the south. In afternoon they strengthen and shift to prevailing
westerlies, extending late into the night (Bradley et al., 2010). Note, WCS seep emissions require winds to "probe or
scan" across the seep field, and thus miss the strong afternoon prevailing winds when emissions are higher.
Specifically, higher wind speeds increase sea-air gas emissions of dissolved near-surface gases (Nightingale et al.,
2000) and increase emissions from higher hydrostatic pressure fluctuation driven by wave height (Leifer & Boles,
2005). Given that prevailing winds are westerlies, higher afternoon emissions will generally (but not always) drift
eastwards, missing WCS.

The diurnal wind pattern from the seep field direction is different from the overall (direction-independent) diurnal
pattern. Typical nocturnal winds are quite weak, 1.5–1.7 m s$^{-1}$ (**Fig. 6**). The strongest diurnal wind change was from
late night to morning: a 20% decrease. Onshore winds (seep direction) in the middle of the night are from synoptic
systems and were associated with the highest $C'$. Winds increase by a few percent to an early afternoon peak,
decreasing through early evening before rising into the night.

The $C$ seep direction trend followed the diurnal wind cycle, increasing by ~20 ppb peaking ~2 hours later in the day
than winds (15:00 versus 13:00 for $C$ compared to $u$, respectively), which likely reflects wave height (which lags wind
strengthening due to the time for wave development) and transport time. Based on sensitivity studies, the diurnal
cycles in $u$ and $C$ correspond to variations of ~7% and ~9% in $E_A$.

Although efforts were made to characterize the diurnal cycle from WCS data, WCS data poorly samples the seep field
for the higher wind speeds that occur in the afternoon which primarily are westerlies. Note, non-linearity in sea-air
evasion with $u$ means the model use of average $u$ underestimates $E_A$. Thus, the contribution of the prevailing afternoon





804 winds to diurnal emissions is significantly underestimated from WCS data. It is worth noting, though, that this factor

805 only affects 25-33% of diurnal emissions. As the true diurnal cycle cannot be derived from WCS data, field data of

806 repeat transects spanning the different phases of a diurnal cycle are needed.


808 **4.7 Future needs and improvements**

809 The sensitivity studies identified areas for improvement and data gaps. These are described in brief below and in more

810 detail in **Supp. Sec. S8**. The largest uncertainty was with regards to partitioning between the inshore and offshore seep

811 trends, which could be determined by a second air quality station, preferably including speciation such as by CEAS

812 analyzers of $CH_4$ and $C_2H_6$. Another important sensitivity was to boundary layer height, $BL$, which varies diurnally

813 and seasonally (Dorman & Winant, 2000) and could be derived from ceilometer data (Münkel, 2007). Another concern

814 is afternoon emissions when seep field emissions bypass WCS, which could be addressed by field work and a second

815 air quality station at a different downwind direction from the seep field.

816

817 The model was limited by available workstation power; however, additional computation power could open

818 improvements such as simulating a range of wind speeds. Mapping offshore wind veering in the seep field would open

819 simulations to provide insights at the seep area scale. Further simulations could add grid cells for evasion

820 corresponding to the downcurrent plumes to assess their contribution.

821

822 **5 Conclusions**

823 In this study, data from an onshore air quality station located downwind of a large marine seep field was analyzed to

824 derive the three-decade-averaged seep field emissions using an inversion model. The modeled emissions were similar

825 to reported emissions; however, this was coincidental given that prior reported emissions were during a period of field

826 quiescence. Highlighting the significance of the COP seep field, ethane and propane emissions suggest the COP seep

827 field contributes 0.04% and 0.12% of the global seep budget, respectively. As a result, COP seep field emissions of

828 19 Tg $CH_4$ yr$^{-1}$ are consistent with global geo-gas budgets of 45 Tg yr$^{-1}$, but inconsistent with significantly lower

829 emissions estimated from ice core isotopic data. Additionally, the approach could be adapted to air quality station data

830 for other sources including terrestrial seeps, production fields, etc., if the sources are spatially constrained.


832 **Data availability.** All data needed to evaluate the conclusions in the paper are present in the paper and/or the
833 Supplementary Materials and/or were submitted to the Mendeley Data Repository, see Leifer, Ira (2020),
834 "Seep_Air_Data", Mendeley Data, V1, http://dx.doi.org/10.17632/znzhxkftm8.1

836 **Supplement.** The supplement contains additional supporting figures and details to complement the manuscript and

837 an interactive map file as a Google Earth archive of the offshore survey data that are presented in **Fig. 2**.






**Author Contributions.** IL Developed and conducted the study, analysed data, and wrote the manuscript. CM analysed
data and edited the manuscript. DB analysed air sample data and edited the manuscript.

**Competing interests.** The authors declare that they have no conflict of interest.

**Acknowledgements.** We would like to gratefully acknowledge the SBCAPCD for providing data from their ongoing
monitoring program and the contribution of Marc Moritsch and Joel S. Cordes in particular for help with these data,
and Doug Wilson for the processed sonar data. We recognize the skill and participation of vessel captains Jeff Wright
and Tony Vultaggio and editorial review by Charlotte Marston, Bubbleology Research International.

**Financial Support.** This work was supported by Plains All American Pipeline and the Bubbleology Research
International, Internal Research and Development (IRAD) fund.



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



**Table of Nomenclature**

| | | |
|---|---|---|
| 1188 | **Table of Nomenclature** | |
| 1189 | $C_{ave}(\theta)$ | Wind direction-resolved average concentration |
| 1190 | $C_{CH4}$ | Methane concentration |
| 1191 | $C_{max}(\theta)$ | Wind direction-resolved maximum concentration |
| 1192 | $C_{med}(\theta)$ | Wind direction-resolved median concentration |
| 1193 | $C'_{Obs}(\theta)$ | Wind direction-resolved WCS observed concentration |
| 1194 | $C'_{Sim}(\theta)$ | Wind direction-resolved WCS simulated concentration |
| 1195 | $u_{ave}(\theta)$ | Wind direction-resolved average $u$ |
| 1196 | $u_{ave}(\theta)$ | Wind direction-resolved average $u$ |
| 1197 | $u_{max}(\theta)$ | Wind direction-resolved maximum $u$ |
| 1198 | $u_{med}(\theta)$ | Wind direction-resolved median $u$ |
| 1199 | $C$ | Concentration |
| 1200 | $C(t,\theta)$ | Wind direction and time-resolved average concentration |
| 1201 | $C'(i,j)$ | Grid cell $i, j$ plume concentration |
| 1202 | $C'(\theta)$ | Wind direction-resolved plume (anomaly) concentration |
| 1203 | $C(\theta)$ | Wind direction-resolved concentration |
| 1204 | $E_A(i,j)$ | Grid cell $i, j$ atmospheric emissions |
| 1205 | $E_A(\theta)$ | Wind direction-resolved atmospheric emissions |
| 1206 | $E_B$ | Seabed (bottom) emissions |
| 1207 | $E_W$ | Emissions to the water column in the near field |
| 1208 | $i$ | Grid cell longitude index |
| 1209 | $j$ | Grid cell latitude index |
| 1210 | $K(r,\theta)$ | Wind direction and distance-resolved correction function to emissions |
| 1211 | $K(\theta)$ | Wind direction-resolved correction function to emissions |
| 1212 | $r$ | Distance from WCS to cell $i, j$ |
| 1213 | $R^2$ | Correlation coefficient |
| 1214 | $t$ | Time |
| 1215 | $u$ | Wind speed |
| 1216 | $u(\theta)$ | Wind direction-resolved wind speed |
| 1217 | $\delta\theta$ | Model wind direction resolution |
| 1218 | $\phi(C)$ | Concentration probability distribution |
| 1219 | $\phi(u)$ | Wind probability distribution |
| 1220 | $\phi(\theta,C)$ | Wind direction and concentration-resolved probability distribution |
| 1221 | $\phi(\theta,u)$ | Wind direction and wind speed-resolved probability distribution |
| 1222 | $\phi(\omega)$ | Sonar return probability distribution |
| 1223 | $\phi_n(\omega)$ | Sonar return probability |
| 1224 | $\phi_n(E_A)$ | Atmospheric emissions probability |
| 1225 | $\theta$ | Wind direction |
| 1226 | $\omega$ | Sonar return |
| 1227 | $\omega(i,j)$ | Grid cell $i,j$ sonar return |
| 1228 | $\psi$ | Wind veering |
| 1229 | $\zeta$ | Relative inshore and offshore emissions |
| 1230 | | |





**Table 1. Atmospheric plume composition and model atmospheric emissions.**

| Gas | Fraction (%) | Emissions (m$^3$ dy$^{-1}$) | Emissions (Mg yr$^{-1}$) |
|---|---|---|---|
| $CH_4$ | 88.5 | 73,900 | 19,300 |
| $C_2H_6$ | 3.1 | 2,590 | 1270 |
| $C_3H_8$ | 4.2 | 3,510 | 2520 |
| $C_4H_{10}$ | 2.76 | 2,300 | 2180 |
| $C_5H_{12}$ | 1.11 | 930 | 1090 |
| $C_6H_{14}$ | 0.13 | 110 | 150 |
| $C_6H_6$ | $4.4 \times 10^{-5}$ | 4.0 | 4.7 |
| $C_7H_{16}$ | 0.04 | 33 | 55 |
| $C_7H_8$ | $1.0 \times 10^{-5}$ | 1.0 | 1.3 |
| NMHC | 11.3 | 9470 | 7280 |
| | | | |
| $C_1$-$C_7$[*] | 85 | 83,400 | 26,600 |
| $CO_2$ | 15 | 18,000 | 12,900 |

* $C_1$-$C_7$ = THC




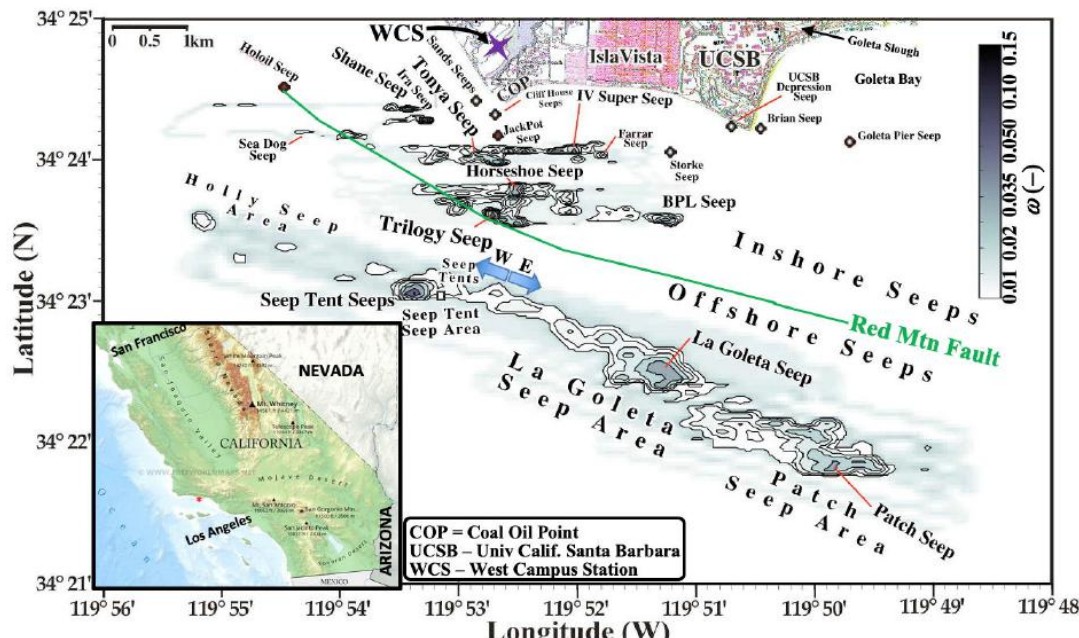

**Figure 1:** Sonar return, $\omega$, map after Leifer et al. (2010). Purple star marks West Campus Station (WCS). Seep names are informal
(**Table S3**), font size corresponds to strength. E-W arrow segregates east and west offshore seepage. Data keys on panels. Inset
shows S. California, red dot marks COP seep field. California inset map from Freeworldmaps (2020).



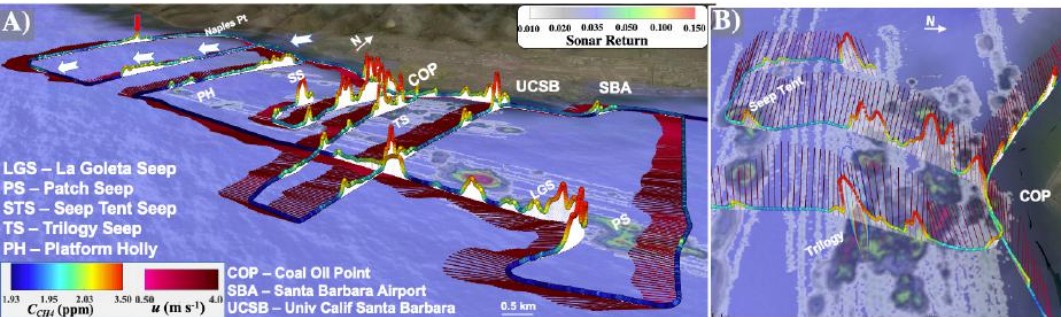

**Figure 2: A)** COP seep field methane, $C_{CH4}$, and winds, $u$, data for 28 May 2016. White arrows show canyon offshore flow. Red arrows show unmapped seepage to the west of the COP seep field. **B)** $C_{CH4}$ and $u$ showing Gaussian plume model for Trilogy Seep. Sonar return, $\omega$, map in background. Data key and seep name key on panel. Displayed in Google Earth environment. **© Google Earth.** (See **Supp. Fig. S6** for overhead view).






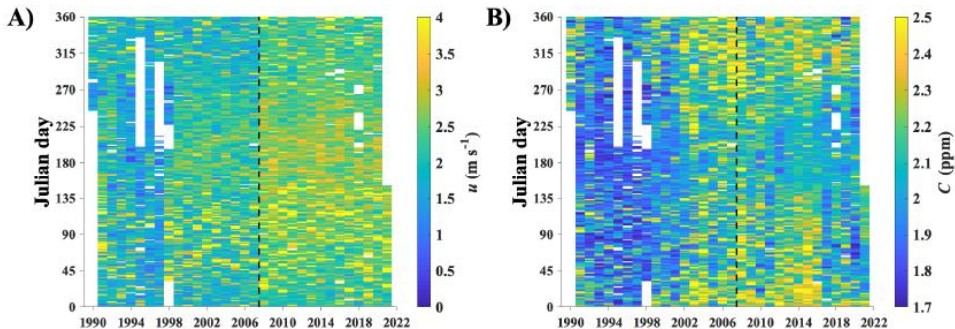

**Figure 3: A)** Daily mean wind speed, $u$, and **B)** total hydrocarbon concentration, $C$. Data key on figure. WCS upgrade on Jan 2008
is shown by a dashed black line. **Supp. Fig. S4** shows raw dataset.






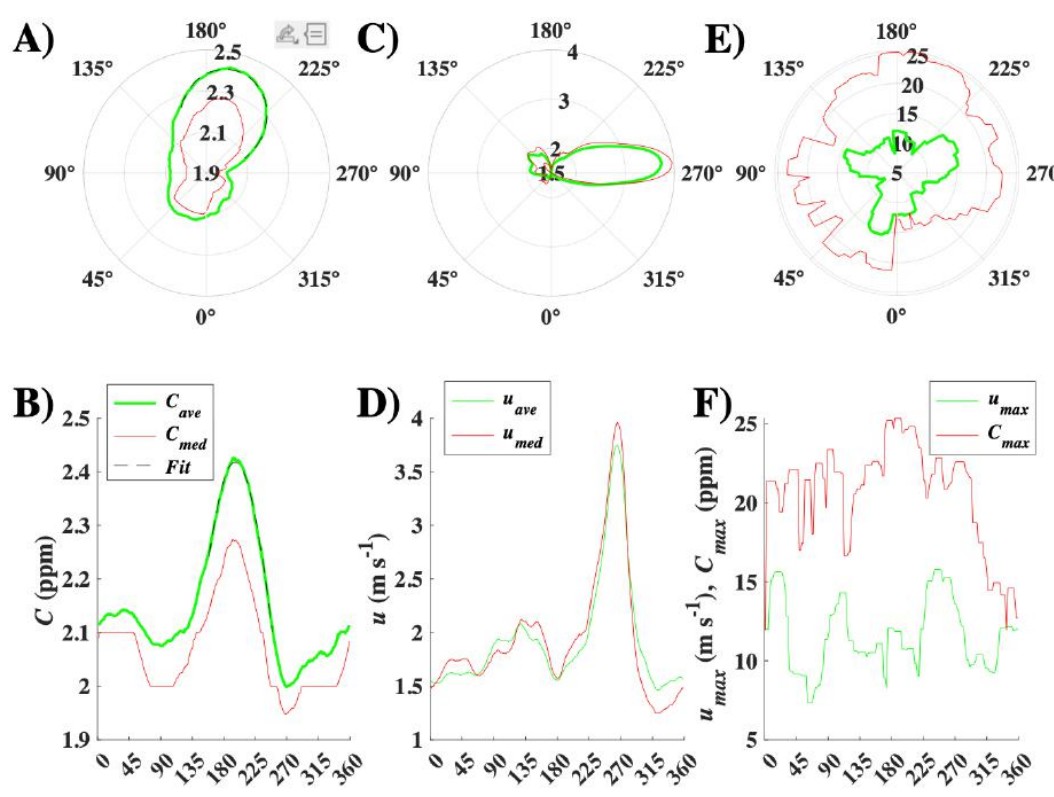

**Figure 4: A, B)** Concentration, $C$, versus wind-direction, $\theta$, 1990-2021 for average, $\boldsymbol{C_{ave}}(\boldsymbol{\theta})$, and median, $\boldsymbol{C_{med}}(\boldsymbol{\theta})$, and fit to
$\boldsymbol{C_{ave}}(\boldsymbol{\theta})$ for $155<\theta<250°$. Data key on panel B. **C, D)** Wind speed, $u$, average, $\boldsymbol{u_{ave}}(\boldsymbol{\theta})$, and median, $\boldsymbol{u_{med}}(\boldsymbol{\theta})$, Data key on panel
D. and **E, F)** Maximum $C$, $\boldsymbol{C_{max}}(\boldsymbol{\theta})$, and wind speed, $\boldsymbol{u_{max}}(\boldsymbol{\theta})$. Data key on panel F. Polar plot oriented as at WCS facing the
COP seep field.





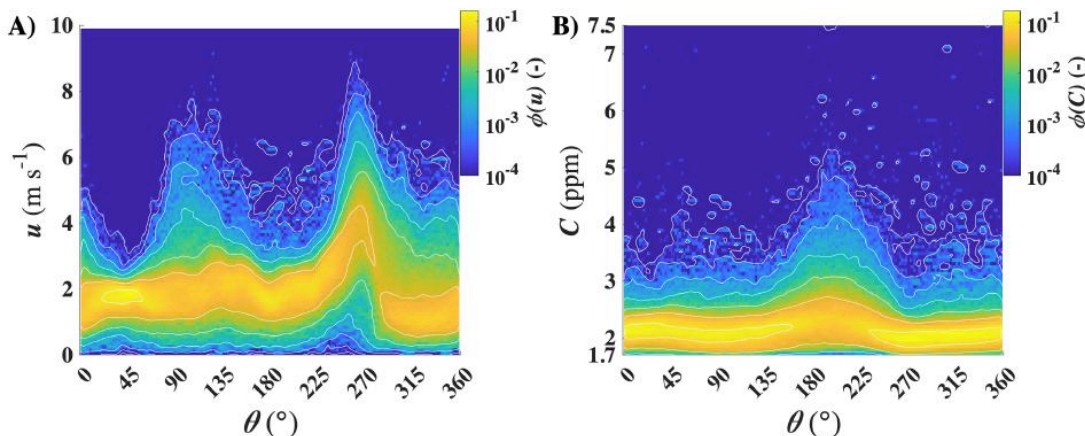

**Figure 5: A)** Wind-direction ($\theta$) resolved wind-speed, $u$, probability distribution, $\phi(\theta, u)$ and **B)** Concentration probability distribution, $\phi(\theta, C)$, for 1990-2016. White dashed line shows edges of seep field. Data key on figure.






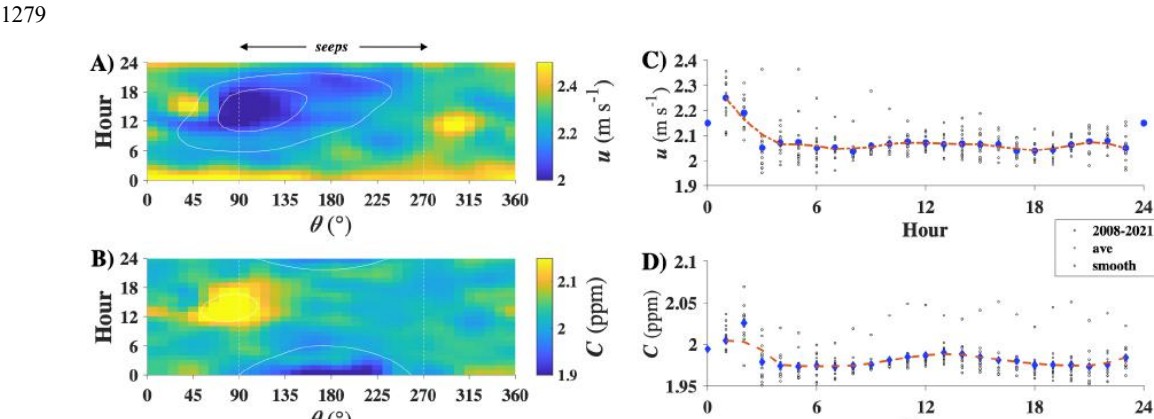

**Figure 6: A)** 2008-2021 averaged wind direction, $\theta$, and hourly-resolved wind speed, $u$, and **B)** concentration, $C$. **C)** Seep-direction
(90–270°), hourly-averaged wind speed, $u$, and **D)** concentration, $C$, averaged, individual years, and 3-year smoothed. Data key on
figure. Midnight data missing due to daily calibration.





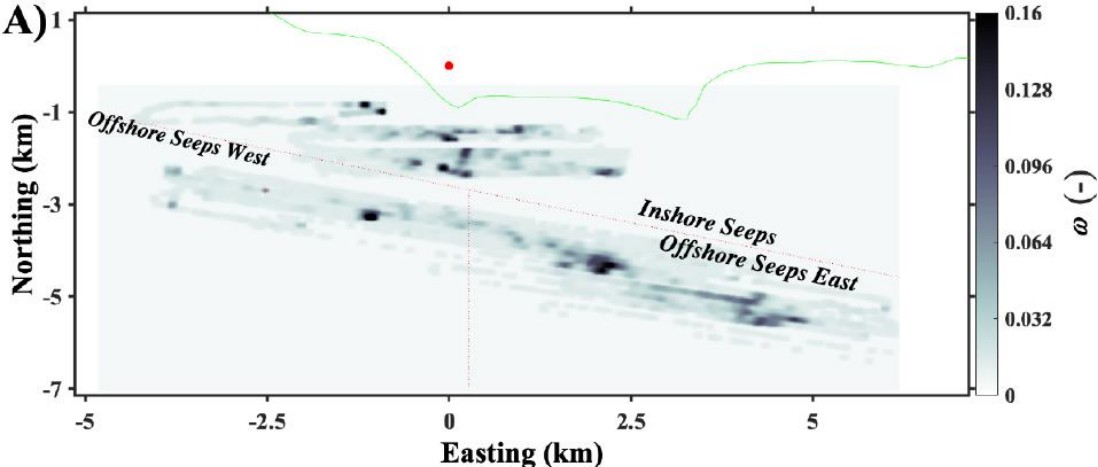

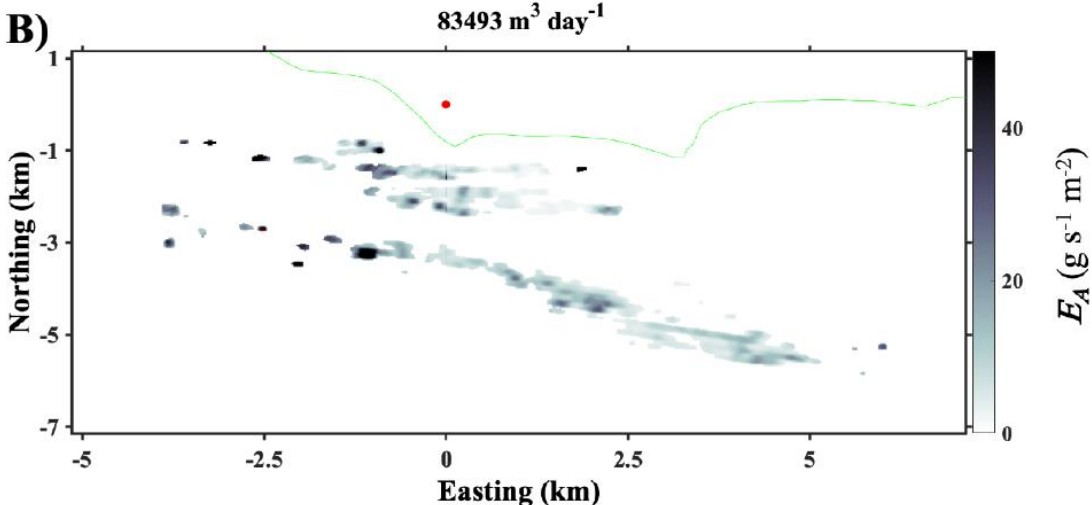

**Figure 7: A)** Sonar return, $\omega$, gridded at 22-m resolution. **B)** Atmospheric emissions, $E_A$. West campus station (red dot) is at
coordinate system origin. Green line is coast line.

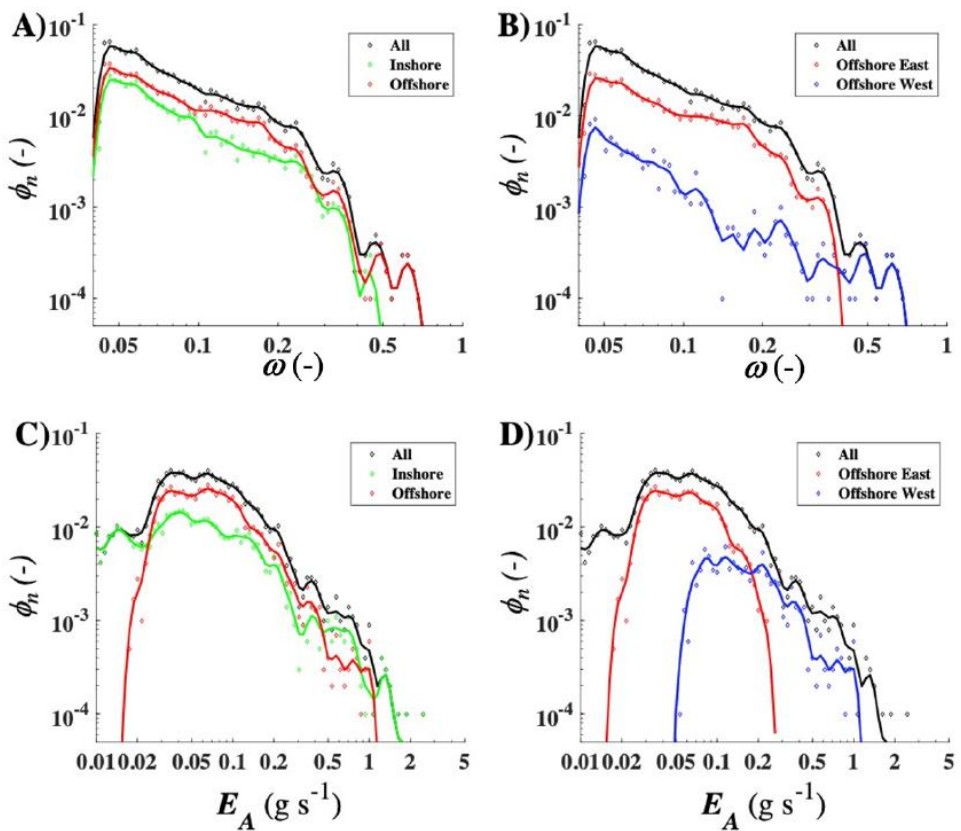

**Figure 8: A)** Sonar return, $\omega$, occurrence probability, $\phi_n(\omega)$, for all seepage, inshore and offshore seepage and **B)** all seepage,
offshore east seepage, and offshore west seepage. **C)** Atmospheric emission, $E_A$, occurrence probability, $\phi_n(E_A)$, for all seepage,
inshore and offshore seepage and **D)** all seepage, offshore east seepage, and offshore west seepage. Data key on panels.






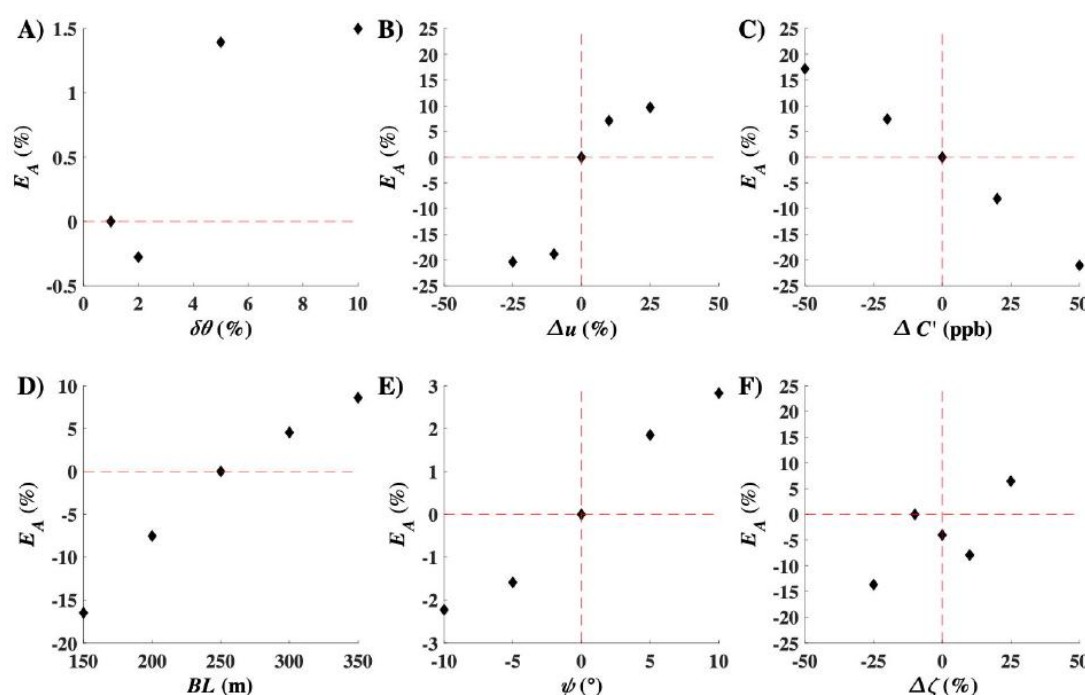

**Figure 9:** Emissions, $E_A$, sensitivity to uncertainty in **A)** model angular resolution, $\delta\theta$, **B)** wind speed variation, $\Delta u$, **C)**
concentration anomaly variation, $\Delta C'$, **D)** boundary layer thickness, $BL$, **E)** wind veering, $\psi$, and **F)** inshore/offshore partition
variation, $\Delta\zeta$. Note different units on different plots. See text for details.





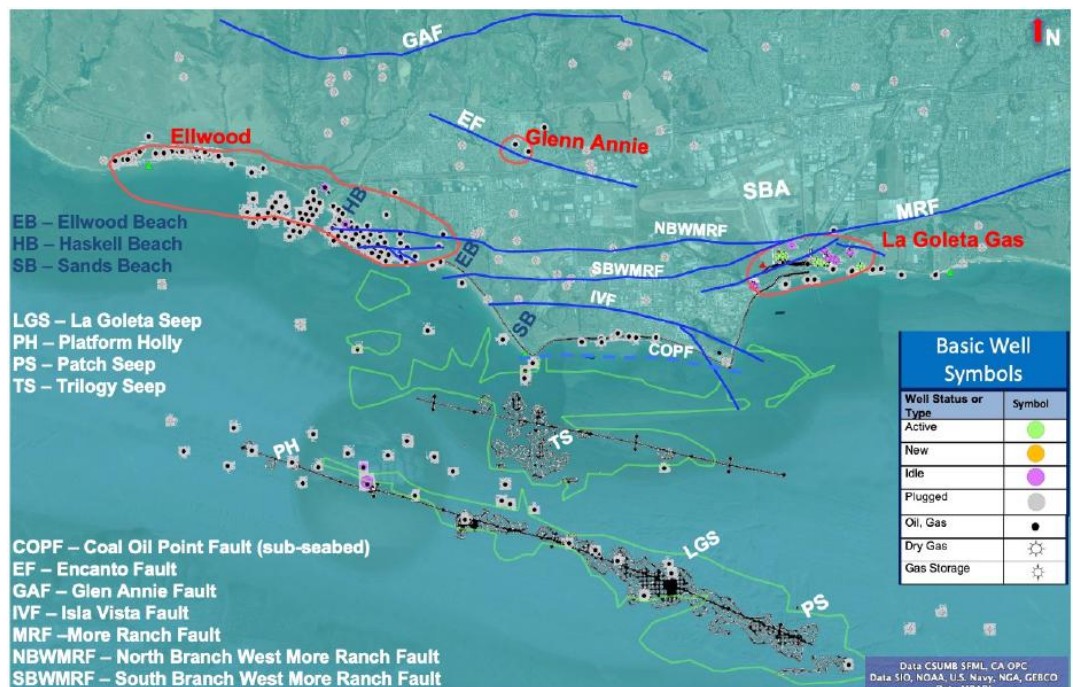

**Figure 10:** Map of the Goleta Plains oil and gas fields, wells, and the Coal Oil Point (COP) seep field. Grey hatch shows 1995
field extent, green outlines the 1940 field extent is from Leifer (2019). Field locations from Olson (1983). Well data from CDOGGR
(2018). Faults from Minor et al. (2009). Seep names are informal. Data keys on panels. Shown in the © Google Earth environment.