# Peer review of "Long-Term Atmospheric Emissions for the Coal Oil Point"

_Atmospheric Chemistry and Physics, 2020_

## Author Comment (AC1)

Review 1

General comments**:** This paper, written by a team led by someone with considerable experience in this field, concerns natural emissions to the atmosphere of methane and other hydrocarbons. As methane is a more potent greenhouse gas than carbon dioxide, it is important that studies providing detailed evidence of natural methane emissions are made available to the scientific community. ACP is clearly a suitable vehicle for this paper.

The paper describes a method of long-term monitoring emissions to the atmosphere of methane derived from natural petroleum reservoirs in shallow waters off the coast of California. This is a valuable method, and could be applied to other natural or anthropogenic methane sources. The Introduction also provides a detailed explanation/ description of natural seabed gas seeps in general, citing numerous relevant sources. Similarly, the fate of methane released from the seabed is also discussed. Whilst this provides valuable context, this material is covered by numerous other papers. There is a danger of the 'context' overwhelming the detail of the method, its application and the acquired results.

Whereas the authors have endeavored to integrate all aspects of a complex study, the paper does not flow well; there are several inconsistencies and other shortcomings that would benefit from more detailed editing.

**We agree that the introductory section flow could be improved. To improve the flow, we have moved the approaches summary from the introduction to a new overview section in Methods. We also agree that as the fate and details of how the trace gases cross the water column are not particularly germane to the findings, and thus, Section 1.3 has been deleted with a few relevant citations and sentences added to Section 1.2. OTOH, the results section does seem to flow:**

1. **Field study focus on individual seep areas, which also introduces the reader to the seep field spatial distribution.**
2. **WCS concentration and wind data and time trends – seasonal and interannual.**
3. **WCS data patterns related to direction (spatial distribution)**
4. **WCS data diurnal patterns which discusses how diurnal time and direction trends are not independent.**
5. **Modeled emissions overall field**

6. **Modeled emissions for field sector**
7. **Sensitivity studies to place the model findings in context**
8. **Ellwood Field emissions**

**Although Ellwood Field emissions are somewhat ancillary to the Seep Field emissions; they fell out of WCS analysis, only comprise 16 lines, and are a very interesting discovery (on a regional basis).**

Furthermore, although some important conclusions have been drawn, there is no discussion of long-term temporal emission variations, as might be expected from the title.

**We have changed the title as the emissions trends are beyond this study. We are well advances on a separate manuscript that investigates these trends; however, adding this work would expand the manuscript unreasonably. We can note that emissions are cyclical with a multi-decadal time scale.**

**Specific comments:**

Lines 138-9: It should be pointed out that, whilst the bubbles from the COP seeps may be oil coated, this is not the case in many (?most) other seeps areas worldwide.

**Fair point. Added.**

Line 212 - Section 2.1: Would it not be appropriate to identify the type(s) of equipment used to acquire these data (especially the THC)?

**Added for winds (MetOne anemometer) and requested from the regulatory agency for THC; however, it will be late-August before this information is available.**

Section 4.2.2 Methane and non-methane hydrocarbon emissions. This section is particularly interesting in the light of the implication by Hmiel et al (2020) that pre-industrial natural geological contributions to atmospheric methane are practically insignificant. Can any comment be made about emission trends over the three decades covered by the data reported here?

**We have changed the title, removing long term as the trends are beyond this study. We are working on a separate manuscript that goes into these trends. Presenting those findings would double the length of this manuscript, which seems inappropriate. We can state that emissions are clearly cyclical with a multi-decadal time scale.**

It would be interesting to compare emission trends with petroleum production - is reservoir depletion reflected by a reduction in seepage emissions? Such a trend is mentioned (Lines 755-778) but only for the Seep Tent Seep.

**See above comment. It appears that since production from the western portions of the field ceased, emissions have been above trend.**

**Technical corrections:**

Line 146: ".... with dissolved plume concentrations decreasing with time ....". Time or distance - or both?

**Both**

Line 151: what is "water-side turbulence"?

**Clarified to "Turbulence in the water boundary layer"**

Lines 173-4: "COP seep field sources from the South Ellwood oil field whose primary source rock is Monterey Formation, which is immature to marginally mature." This could be re-cast as "The source of the methane of the COP seep field is the South Ellwood oil field, which contains petroleum from the immature to marginally mature Monterey Formation."

**Rewritten for clarity as suggested. Also rewrote the next sentence.**

Line 191: wet season (singular).

**Yes. Thanks.**

Line 230: There is no verb in this sentence.

**Verb added.**

Line 250: "The plume inversion model is a three-step process". Surely this should be "The plume inversion modelling is a three-step process".

**Rewritten The plume inversion is a three-step process.**

Line 252: What is C'?

*C'* **is the concentration anomaly. Defined.**

Line 255: "is fit" should be "was fitted" - although present and past tense seem to be interchangeable in this section.

**As the model is in use in many studies and it is now described in present tense.**

Line 328: For the benefit of readers, it should be noted that the 'Seep Tent' refers to an installation whereby the natural seabed seepage was captured and utilized along with gas produced from nearby petroleum fields. Suggest moving text from Lines 765-769.

**This is a good suggestion, done! Also got rid of another sentence that was duplicated in the process.**

Line 367: "almost due south to the". Surely WCS is almost due north of Coal Oil Point?

**Yes. Fixed.**

Line 373: "allowed far higher values of C and u" - add " to be measured"

**Thanks.**

Line 437: "C and u for the seep field direction, useep, and Cseep, respectively" should be: C and u for the seep field direction, and Cseep, and useep , respectively.

**Thanks, swapped.**

Line 450: replace "largely" with "mainly"

**Largely on line 458 deleted, largely on line 447 changed to primarily.**

Line 579: "A range of approaches are available" - a range … is available!

**Rewritten and corrected from CH₄ seepage to gas seepage (which is what is generally reported).**

Lines 665-680 is clumsily worded, and units are mixed (tons - should be tonnes and could be abbreviated to 't' - Gg, Mg and nmol). All previous multi-authored papers are cited using multiple names, except Römer et al. 2017. e.g. Line 666-8: "e.g., summary Römer et al. (2017) where emissions for 12 different seep areas including for sites in the North Sea, Pacific north west, Gulf of Mexico, etc., were 2-480 tons yr-1, multiple orders of magnitude less than seabed emissions for Coal Oil Point. Römer". Poorly worded. Suggest: 'For example, Römer et al. (2017) identified emissions from 12 different seep areas (in the North Sea, Pacific north west, Gulf of Mexico, etc.) of 2 to 480 tonnes yr-1.' [N.B. the last 12 word duplicate the previous sentence. Römer et al. likely used metric tonnes rather than US (Imperial) tons - why not abbreviate to 't'?]

**This section has been rewritten as we agree it was clumsy. We also added the detail that in Romer et al. (2017) the range of seep emissions are for seabed emissions. We also separated the paragraph between indirect estimates of atmospheric emissions (based on seabed measurements) and direct estimates of atmospheric emissions.**

**WRT the citation, Schmale et al. (2010), this was an endnote error, and since endnote normally works, it was overlooked. It has been corrected.**

**Thanks for the correction on units, I have decided to use Mg which has no confusion with American short ton and long ton, and all the US unit backwardsness.**

Line 670: Tommelieten should by spelled Tommeliten.

**Thanks, Fixed.**

Line 693: "emissions were" estimated as …

**Fixed. Thanks.**

Section 4.4: to conform to section 4.2.2, emissions should quantified by mass rather than volume.

**Here we use volume as that is how Clark et al. report them, which is now noted.**

Line 771: If the Seep Tent Seep is post 1978, how come it was observed in 1970? (Line 768).

**This was a typo and should have been 1970 as you noted.  The next line about a sea boil was duplicated and is now deleted.**

Line 783: "WCS seep emissions" - surely you mean measurements, not emissions.

**Revised to "Note, measurement of WCS seep emissions"**

Line 1199: If C is defined, then C' should also be defined.

**Agreed.**

Fig. 4: Why are the rose diagrams plotted with 0 (presumably representing North) at the bottom, South at the top, and therefore East and West reversed from their intuitive places? Also, it would make interpretation of the rose diagrams easier if they were superimposed on the map; this would enable correlations with seep locations more intuitive

**We wanted to show the wind rose as if one were standing at WCS looking towards the seep field (the south). Showing all three wind roses on three maps would be unwieldy.**

**We will break the figure into two figures with the concentration wind rose superimposed on a map and the other two wind roses on the side.**

---

## Author Comment (AC2)

We would like to take this opportunity to respond to reviewer 2, who both raised some good points, which were addressed through improvements to the manuscript, and some points that do not apply to our study and paper (as written).

**Comment on manuscript length:**
We agree that the introductory and methods section could be organized better, as also commented constructively by reviewer 1, and for which we have shortened, re-organized including removing duplicative and ancillary material. Please see response to reviewer 1. That said, we leave supplemental material as supplemental to (as requested by reviewer 2) not lengthen the manuscript.

**Comment on novelty**
We never argued that the approach is novel - we wrote (and maintain) that it is the application to marine seepage that is novel.

**Comment on THC measurement details missing**
The measurements were made by a regulatory agency and as a result, it should be unnecessary to have the measurement details for the purpose of manuscript evaluation. When modelers use NOAA data they do not report on NOAA measurement approaches. Still, we contacted the agency to find the instrumentation details and added them to the manuscript.

**Comment on uncertainty and sensitivity**
We did an extremely comprehensive series of sensitivity studies because the preferred approach to assess uncertainty – Monte Carlo – is computationally impossible with the resources (workstations) available and an uncertainty approach such as quadrature is inappropriate. Specifically, uncertainty arises from several phenomena for which data are unavailable, such as wind veering and the boundary layer height, and for which both data and processes are unknown such as the emissions ratio between inshore and offshore seepage trends. Quadrature implies that the functional relationship between emissions and the many parameters that affect it can be expressed by an equation – the reality is that for this system, it cannot. The suggestion to run the model with the upper values at 1 is incorrect for assessing uncertainty – it would calculate the upper limit of overall error. To assess uncertainty - a Monte Carlo approach is needed.

We note, that the fact that overall emissions agree reasonably well with published values provides confidence that uncertainty is not so large as to make the model useless.

**Comment on using reanalysis wind and boundary layer product in the inversion**
Our experience of these products is that they fail to capture the necessary fine-scale wind structure on sub-kilometer length scales (clearly evident in the presented data, but perhaps overlooked). In mountainous coastal terrain, such as for California and the study area in particular, Hysplit tends to predict wind trajectories that run right through the coastal mountains. As such these products are inappropriate for the analysis used in this study.

**Comment on not knowing how background THC concentration was derived.**

The derivation of background concentration was provided in the text.

**Comment on potential for biases and errors over time invalidating the effort.**
We did the best we could with available information and data and derived similar values to those reported from snapshot field surveys. Moreover, in the field, annualized budgets are regularly derived from single surveys and assumed to represent emissions even over decades. For example, *Hornafius et al*. (1999) is still cited decades later despite being a 1995 snapshot. Thus, in this regards, an analysis based on 30 years of data obviously improves on current state of knowledge.

Moreover, there is a basis for not expecting dramatic seep field changes on less than geological timescales - geological systems change slowly – with emissions highly spatially constrained by faults and fractures that are "fixed" in solid rock. Specifically, migration from below that charges the shallow reservoir must balance over long times with emissions to the seabed through fractures above. Moreover, as studied in the sensitivity studies, overall emissions are weakly sensitive to shifts in the location of active seepage, which can change on short time scales, even to shifts between the offshore and onshore trends and for along-trend shifts, which were investigated in the wind-veering sensitivity studies.

**Comment on evasion process is complex and not captured by the model.**
We direct the reviewer and reader to the manuscript where we presented data that the highly soluble gas, $CO_2$, is in similar ratios in the atmospheric plume and the seabed. This demonstrates that gases that dissolved during bubble rise are transported by the upwelling flow to the sea surface and then evade into the atmosphere. This upwelling process was detailed in the introductory material and in the discussion.

Finally, the reviewer may not have understood how the model functioned. Specifically, overall emissions are not affected whether hydrocarbons evade in the grid cell above the seabed or several grid cells downcurrent – their contribution is counted. Thus, the elevated concentration of seep gas that is observed from a specific direction is assigned to the nearest sources in the sonar map. The model does not consider whether a molecule of methane came from the atmospheric bubble plume or sea surface evasion – a molecule of methane is a molecule of methane – the model simply derives the emissions for the measurement.

Only the fraction that evades from downcurrent of the seep field sonar extent remains unaccounted. This was noted as needed future work.

**Comment on THC not being a conserved tracer.**
Given that the transport time at typical wind speeds is just 20-30 minutes, THC, whose composition was reported, is conserved - the NMHC is predominantly light alkanes with lifespans of weeks to longer. We now note that chemical changes are unlikely on the transport time for completeness.

**Comment that the model assumes constant and uniform emissions.**

Our model does not make this assumption, and thus it is unclear as to how this comment arises.

**Comment that unknown sea-surface state using the partitioning calculate the seabed emissions using a 50:50 partitioning.**

We note that the study goal was to assess the atmospheric emissions NOT to assess seabed emissions. We do so solely to allow comparison with Hornafius et al. (1999). To enable a comparison we used the *Clark et al*. (2000) partitioning, which was based on a large field study. This partitioning is very weakly related to sea-surface state (outside storms) – only to the fraction that dissolves below the wave-mixed layer – dissolved gases above the wave-mixed layer evade sooner (tens of meters) or later (kilometers), and thus were considered by *Clark et al*. (2000) in their partitioning.

**Comment on inability to separate terrestrial and marine sources and emissions**

We agree that for an air quality station in Los Angeles trying to deconvolve the emissions from for example, La Brea, our approach might be ineffective; however, the COP seep field is NOT in Los Angeles. It is offshore. And upwind for prevailing airflow patterns is the vast Pacific. As such the fact that the $C(\theta)$ distribution closely matches the seep field extent demonstrates the validity of the approach to segregate terrestrial and marine sources – distant sources create very broad features, and present as a gradient across the seep field.

**Comment on the offshore validation study as unconvincing.**

The purpose of the validation study was not to compare with WCS but rather to show that the offshore sources are well-represented by Gaussian plumes, a key assumption.

---

## Author Response (AR2)

Thank you for addressing the reviewers comments and the heavily revised manuscript. I am recommending the manuscript be published following your attention to the following minor points:

1) L14 of the abstract notes that the plume air THC was 85% CH4 and 20% CO2. Is this correct? This exceeds 100% and I wouldn't have thought to characterize CO2 as THC? Perhaps the sentence as written has me confused (but could also confuse a novice reader). Table 1 lists CO2 as 15%. I follow the logic of table 1, but the wording of the abstract is not clear. Using table 1, CH4 is 88.5% of THC and 85% of total carbon on a molar basis, correct?

**Although this was clear to me when I wrote it, my re-read also was confusing, so I went back to the data and recalculated and decided for clarity to add a fourth column to Table 1 – fraction of total carbon, defined as THC+CO2+CO, and now define these clearly in the abstract, table, and text.**

2) L17 of the abstract has emissions in units of M3 THC day-1. To the layperson this unit is confusing, as most are thinking in terms of a mass/unit time rather than a volume per unit time. Again, I follow the units of table 1 (far column), but the abstract units of volume/time are hard to think about. Also, the abbreviation for day-1 should probably be d-1

While these units may be standard for one community, other adjacent fields are often thinking of emissions to the atmosphere as molecules / (area x time) or some equivalent. It would be helpful to these readers if the abstract could make this connection.

**We have added (in parentheses) that emissions are (27 Gg *THC* yr$^{-1}$ based on 19.6 g mole$^{-1}$ for *THC*), and corrected to d$^{-1}$.**

**In addition, we did a thorough, line by line, careful proofread of the entire manuscript and made a number of minor improvements for clarity, as well as fixing one or two figure call outs, and added a more recent citations to the primary CH$_4$ loss mechanism (Zhao et al (2020)) and a more recent citation on what is seepage (Ciotoli 2020) to buttress the not recent Abrams (2005).**

---

## Author Response (AR3)

11 Oct 2021

Dear Sirs,

I have edited the paper to include the Google Earth copyright mark, and also performed a careful line by line pass through the paper and supplemental material making numerous small grammatical changes for clarity. The track changes version of the supplemental material is available.

Sincerely,

Ira Leifer and co-authors